# Deep Single-Index Fréchet Regression

Muqing Cui [1]   Yidong Zhou [1]   Su I Iao [1]   Hans-Georg Müller [1]

## Abstract

Predicting outputs that are located in non-Euclidean spaces, such as probability distributions, networks, and symmetric positive-definite matrices, is becoming increasingly important in modern data analysis, particularly when inputs are high-dimensional. We propose *DeSI* (Deep Single-Index Fréchet Regression), a semiparametric framework for regression with metric space-valued outputs and multivariate inputs that assumes a single-index structure for the conditional Fréchet mean. DeSI estimates an interpretable index direction, which quantifies the relative importance of inputs, using a deep neural network, and performs Fréchet regression along the resulting one-dimensional index in the target metric space. This structure mitigates the curse of dimensionality while retaining interpretability, which stands in contrast to standard deep neural networks. We establish theoretical guarantees for DeSI, including uniform approximation and convergence rates, and demonstrate its strong predictive performance through simulations on distributions, networks, and symmetric positive-definite matrices, as well as an application to compositional mood data from New Jersey.

## 1. Introduction

Regression with non-Euclidean outputs has become increasingly important as modern data analyses routinely involve structured responses such as probability distributions (Petersen et al., 2022), networks (Zhou & Müller, 2022), symmetric positive-definite matrices (Thanwerdas & Pennec, 2023), and compositional data (Scealy & Welsh, 2014). Such data, often referred to as *random objects* (Müller, 2016), typically reside in general metric spaces and do

not admit standard algebraic operations such as addition or scalar multiplication, limiting the applicability of classical regression methods designed for Euclidean outputs.

Recent advances in statistical learning have extended regression methodology beyond Euclidean outputs to responses in general metric spaces. Early approaches relied on Euclidean embeddings (Faraway, 2014) or kernel-based similarity measures (Hein, 2009). *Fréchet regression* (Petersen & Müller, 2019) extends classical regression to metric space-valued outputs; global Fréchet regression extends multiple linear regression and local Fréchet regression generalizes local linear smoothing. Subsequent work expanded the Fréchet regression paradigm to include sufficient dimension reduction (Ying & Yu, 2022; Zhang et al., 2024), principal component regression (Song & Han, 2023), additive regression models (Lin et al., 2023; Han et al., 2020; Jeon & Park, 2020; Jeon & Van Bever, 2025) and tree-based methods (Capitaine et al., 2024; Qiu et al., 2024; Zhou et al., 2025).

Despite these advances, high-dimensional predictors remain a central challenge. Global Fréchet regression can be too restrictive to capture complex regression relationships, while kernel-based local Fréchet regression suffers from the curse of dimensionality when applied directly to multivariate inputs. Dimension reduction is therefore essential for scalable and flexible modeling, particularly in applications where interpretability of predictor effects is desired. Deep neural networks have become a standard tool for learning flexible representations from high-dimensional inputs. However, existing deep learning approaches for metric space-valued outputs (Iao et al., 2025) often rely on multi-stage pipelines, in which representation learning and Fréchet regression are trained separately. As a result, learned representations are not directly optimized for the final Fréchet regression objective, and interpretability is typically limited.

We propose **Deep Single-Index Fréchet Regression (DeSI)**, a semiparametric, *end-to-end* framework for regression with metric space-valued outputs and high-dimensional inputs. DeSI uses a deep neural network to learn a *one-dimensional index* and performs local Fréchet regression along this index in the target metric space. This design mitigates the curse of dimensionality while retaining *interpretability* through an estimated index direction that quantifies the relative importance of predictors. The proposed

---

[1]Department of Statistics, University of California, Davis, CA, United States. Correspondence to: Hans-Georg Müller <hgmueller@ucdavis.edu>.

*Proceedings of the $43^{rd}$ International Conference on Machine Learning*, Seoul, South Korea. PMLR 306, 2026. Copyright 2026 by the author(s).

framework integrates index learning and Fréchet regression within a unified optimization objective, enabling the learned index to be directly tailored to prediction in the target metric space. This coupling allows DeSI to leverage the flexibility of deep neural networks while preserving the statistical structure of single-index Fréchet models.

Our main contributions are summarized as follows:

- We propose DeSI, a deep single-index Fréchet regression framework that jointly learns an index function via a neural network and performs local Fréchet regression for metric space-valued outputs within a unified optimization objective.

- DeSI yields an interpretable index direction that quantifies the relative importance of predictors, combining the representational flexibility of deep learning with the interpretability of single-index models.

- We establish theoretical guarantees for DeSI, including universal approximation of the index function and convergence rates for the resulting estimator.

- Through simulations and real-data applications, we demonstrate that DeSI consistently outperforms existing Fréchet regression and deep learning baselines.

The code implementation of this paper is available at the repository: https://github.com/ChopinMQ/Deep-Single-Index-Frechet-Regression.

### 1.1. Related Work

**Single-index Fréchet regression.**  Single-index Fréchet regression methods project multivariate predictors onto a scalar index and apply local Fréchet regression along the index, reducing effective input dimensionality and providing interpretable direction estimates (Bhattacharjee & Müller, 2023; Ghosal et al., 2023). These approaches rely on linear index functions and computationally intensive optimization procedures, which can limit flexibility and scalability. DeSI generalizes this framework by learning the index using a deep neural network while retaining interpretability.

**Deep learning for metric space-valued regression.**  Several recent methods integrate deep learning with Fréchet regression by learning representations or weights for constructing Fréchet means (Iao et al., 2025; Zhou et al., 2026; Kim et al., 2025). In particular, Deep Fréchet Regression (Iao et al., 2025) employs neural networks to relate Euclidean inputs to a low-dimensional manifold representation of the metric space-valued output; however, it relies on a restrictive low-dimensional manifold assumption and a multistage training pipeline. In contrast, DeSI integrates index learning and local Fréchet regression in a unified end-to-end

framework and explicitly enforces a single-index structure to balance flexibility, scalability, and interpretability.

## 2. Preliminaries

Let $(\Omega, d)$ be a complete and separable metric space, and let $Y$ be a random object taking values in $\Omega$. Assume that $Y$ has finite second moment, that is, there exists $y \in \Omega$ such that $\mathbb{E}\left[d^2(Y, y)\right] < \infty$. Due to the lack of algebraic operations such as addition and scalar multiplication in general metric spaces, the classical definition of expectation is no longer applicable. The *Fréchet mean* extends the definition of expectation to general metric spaces (Fréchet, 1948),

$$\mathbb{E}_\oplus(Y) = \arg\min_{y \in \Omega} \ \mathbb{E}\left[d^2(y, Y)\right].$$

In classical regression with a scalar output $Y \in \mathbb{R}$ and a vector input $X \in \mathbb{R}^p$, the target is the conditional mean $\mathbb{E}(Y \mid X = x)$. When the output instead takes values in a general metric space $\Omega$, a natural extension is the *conditional Fréchet mean* (Petersen & Müller, 2019) $m(x) = \mathbb{E}_\oplus(Y \mid X = x)$, which minimizes the conditional expected squared distance and reduces to the conditional mean in the Euclidean case:

$$\mathbb{E}_\oplus(Y \mid X = x) = \arg\min_{y \in \Omega} M(x, y),$$
$$M(x, y) = \mathbb{E}\left[d^2(y, Y) \mid X = x\right].$$

Since the conditional Fréchet mean $m(x)$ is generally unknown and may depend nonlinearly on $x$, Petersen & Müller (2019) proposed *local Fréchet regression (LFR)* as a nonparametric estimator that generalizes local linear smoothing to metric space-valued outputs. This estimator computes a weighted Fréchet mean of the observed outputs, where the weights are obtained via kernel-based local linear smoothing in the input space and depend on the proximity between each $X_i$ and the query point $x$. This construction directly parallels local linear regression in Euclidean spaces, with squared distances $d^2(\cdot, \cdot)$ replacing squared Euclidean loss.

Formally, given i.i.d. samples $\{(X_i, Y_i)\}_{i=1}^n$, the LFR estimator at $x$ is defined as the minimizer of the weighted Fréchet objective

$$\widetilde{M}_h(x, y) = n^{-1} \sum_{i=1}^n \widetilde{w}_{in}(x, h) \, d^2(Y_i, y), \qquad (1)$$

where the weights are given by $\widetilde{w}_{in}(x, h) = \widetilde{\sigma}_0^{-2} K_h(X_i - x)\{\widetilde{\mu}_2 - \widetilde{\mu}_1(X_i - x)\}$, with $\widetilde{\mu}_j = n^{-1} \sum_{i=1}^n K_h(X_i - x)(X_i - x)^j$, $\widetilde{\sigma}_0 = \widetilde{\mu}_0\widetilde{\mu}_2 - \widetilde{\mu}_1^2$, and $h$ denoting the bandwidth associated with the kernel $K_h(\cdot) = h^{-1}K(\cdot/h)$. Further intuition and technical details are provided in Appendix A.

To illustrate the generality of the proposed DeSI framework, we next introduce several representative metric spaces

that arise in real data applications. These spaces are used throughout our simulation studies and real-data analyses, demonstrating the broad applicability and adaptability of the proposed methodology.

**Example 1** (Symmetric positive-definite matrices). Consider the space of $q \times q$ symmetric positive-definite (SPD) matrices, denoted by $\mathcal{S}_+^q$, which commonly arise as covariance or correlation matrices. Several metrics have been proposed for $\mathcal{S}_+^q$, including the Frobenius, power (Dryden et al., 2009), log-Cholesky (Lin, 2019), and Bures–Wasserstein metrics (Bhatia et al., 2019). We focus on the log-Cholesky metric, which admits a convenient representation via Cholesky decompositions. For $k = 1, 2$, let $S_k = L_k L_k^\top$ be the Cholesky decompositions of $S_k$, where $L_k$ is lower triangular with positive diagonal entries, and write $L_k = D_k T_k$ with $D_k$ diagonal and $T_k$ unit lower triangular. The log-Cholesky distance between $S_1$ and $S_2$ is

$$d_{\mathrm{LC}}(S_1, S_2) = \sqrt{\|\log D_1 - \log D_2\|_F^2 + \|T_1 - T_2\|_F^2}.$$

SPD-valued data and geometry-aware learning methods have been widely studied in machine learning, including both discriminative (Wang et al., 2023) and generative models (Li et al., 2024) defined directly on $\mathcal{S}_+^q$.

**Example 2** (Networks). Consider the space of simple, undirected, weighted networks with a finite number of nodes and bounded edge weights. Each network can be uniquely characterized by its graph Laplacian, which is a symmetric positive semi-definite matrix. Equipped with the Frobenius metric $d_F$, the space of graph Laplacians can be used to characterize the space of networks (Kolaczyk & Csárdi, 2014; Zhou & Müller, 2022; Severn et al., 2022). For instance, graph convolutional networks define convolutional operators using the graph Laplacian, exploiting its spectral properties to encode the intrinsic geometric and relational structure of graph-structured data (Kipf & Welling, 2017). These have proven to be of practical value in fields such as protein design (Notin et al., 2024) and atomistic materials chemistry (Batatia et al., 2025).

**Example 3** (Univariate Probability Distributions). Consider the Wasserstein Space $(\mathcal{W}, d_{\mathcal{W}})$ of univariate probability distributions with finite second moments, equipped with the 2-Wasserstein metric

$$d_{\mathcal{W}}^2(\mu_1, \mu_2) = \int_0^1 \left[ F_{\mu_1}^{-1}(p) - F_{\mu_2}^{-1}(p) \right]^2 \mathrm{d}p,$$

where $F_{\mu_1}^{-1}$ and $F_{\mu_2}^{-1}$ are quantile functions for $\mu_1$ and $\mu_2$ respectively (Villani, 2003; Ambrosio et al., 2008; Panaretos & Zemel, 2020). It has been widely used, including Wasserstein auto-encoders (Kolouri et al., 2019; Tolstikhin et al., 2018) and Wasserstein regression (Chen et al., 2023a;b).

**Example 4** (Compositional Data). Compositional observations encode relative contributions across categories and

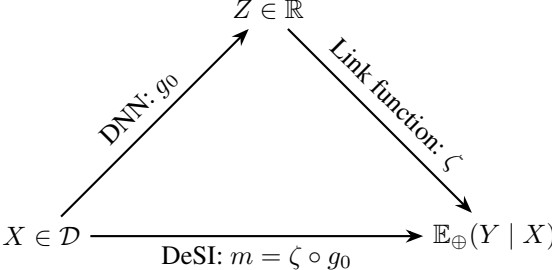

*Figure 1.* Schematic illustration of the Deep Single-Index Fréchet Regression (DeSI) framework. A deep neural network learns a single index $Z = g_0(X)$, which is then mapped through a link function $\zeta$ to estimate the conditional Fréchet mean $\mathbb{E}_\oplus(Y \mid X)$.

are subject to a unit-sum constraint. Such data arise in a variety of settings, including resource allocation across daily activities, normalized attention weights in neural networks, and fractional usage patterns in transportation systems. Let

$$\Delta^{d-1} = \left\{ u \in \mathbb{R}^d : u_j \geq 0, \sum_{j=1}^d u_j = 1 \right\}$$

denote the $(d - 1)$-dimensional unit simplex. A common geometric representation is obtained via the square-root transformation $y = \sqrt{u}$, which maps $\Delta^{d-1}$ onto the positive orthant of the unit sphere $\mathbb{S}_+^{d-1}$ (Scealy & Welsh, 2011; 2014). Under this embedding, the natural geodesic distance between two compositions $y_1, y_2 \in \mathbb{S}_+^{d-1}$ is given by $d_g(y_1, y_2) = \arccos(y_1^\top y_2)$, corresponding to the arc length of the shortest great-circle path connecting the two points.

## 3. Deep Single-Index Fréchet Regression

### 3.1. Problem Formulation

Let $(X, Y)$ be a random pair, where the predictor $X \in \mathbb{R}^p$ has compact support $\mathcal{D}$ and the output $Y$ takes values in a metric space $(\Omega, d)$. Our goal is to estimate the conditional Fréchet mean $m(X) = \mathbb{E}_\oplus(Y \mid X)$. We assume that $m(\cdot)$ admits a *single-index structure*, meaning that there exist a direction vector $\theta_0 \in \mathbb{R}^p$, a scalar index function $g_0(X) = X^\top \theta_0$ and a link function $\zeta : \mathbb{R} \mapsto \Omega$ such that

$$m(X) = \zeta\left(g_0(X)\right).$$

As illustrated in Figure 1, a deep neural network (DNN) is used to estimate the single index $Z = g_0(X)$, which is subsequently mapped to the conditional Fréchet mean via the link function $\zeta$. This assumption reduces the potentially high-dimensional dependence of the conditional Fréchet mean on $X$ to a one-dimensional representation $Z$, while retaining flexibility for modeling non-Euclidean outputs through the link function $\zeta$.

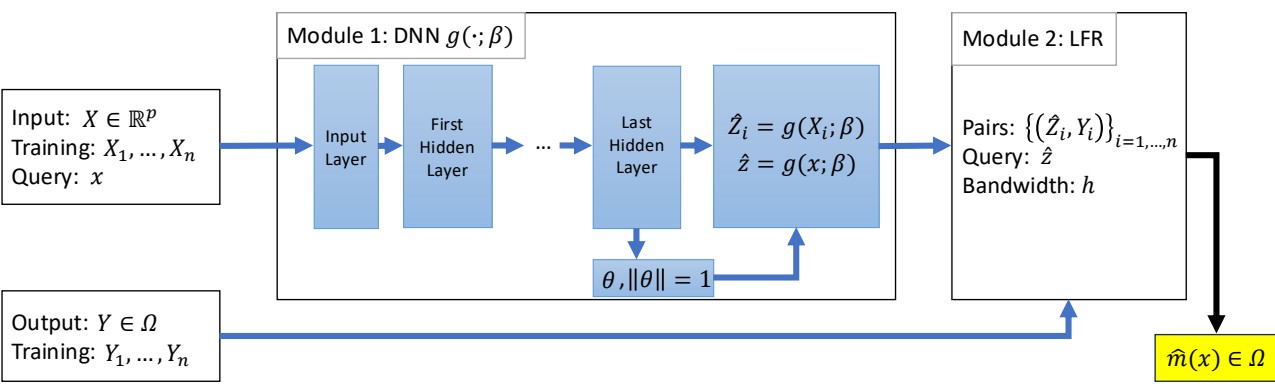

*Figure 2.* Deep Single-Index Fréchet Regression (DeSI) framework. A neural network (DNN module) learns index directions and projects inputs onto a one-dimensional index, followed by LFR (LFR module) with a trainable parameter $h$ as the bandwidth.

The proposed DeSI framework is implemented using two interconnected modules: a DNN for single-index learning and a LFR module for output prediction. The full architecture is depicted in Figure 2. The DNN, denoted by $g(\cdot; \beta)$ with parameters $\beta$, maps each input $X_i$ to a scalar index value $\widehat{Z}_i = g(X_i; \beta)$. Given the projected training pairs $\{(\widehat{Z}_i, Y_i)\}_{i=1}^n$, LFR is then applied along the one-dimensional index to predict the metric space-valued output.

### 3.2. DNN for Single Index Learning

To learn the single-index projection, we employ a deep neural network (DNN) with $L$ hidden layers. The $l$th hidden layer contains $p_l$ neurons, with weight matrix $Q_l \in \mathbb{R}^{p_l \times p_{l-1}}$ and bias vector $b_l \in \mathbb{R}^{p_l}$ for $l = 1, \ldots, L$. Denoting the full collection of network parameters by $\beta = \{Q_l, b_l\}_{l=1}^L$, the DNN defines a composite mapping $g : \mathbb{R}^p \mapsto \mathbb{R}$ given recursively by

$$
\begin{aligned}
g(x) &= x^\top v_L(x)/\|v_L(x)\|, \\
v_L(x) &= \sigma(Q_L v_{L-1}(x) + b_L), \\
&\vdots \\
v_0(x) &= x,
\end{aligned}
\tag{2}
$$

where $p_0 = p_L = p$ and $\sigma(u) = u\mathbf{1}\{u \geq 0\} + \alpha u\mathbf{1}\{u < 0\}$ denotes the leaky rectified linear unit (Leaky ReLU) activation function with $\alpha \in (0, 1)$. Compared with the standard ReLU, Leaky ReLU maintains nonzero gradients for negative inputs, alleviating the dying ReLU problem and leading to more stable optimization (Maas et al., 2013).

The output of the final layer $v_L(x) \in \mathbb{R}^p$ is normalized to unit length and interpreted as a direction vector $\widehat{\theta}(x) = v_L(x)/\|v_L(x)\|$, which induces a scalar index via projection $g(x; \beta) = x^\top \widehat{\theta}(x)$.

Evaluating this mapping at the training samples yields

subject-specific directions $\widehat{\theta}_i = \widehat{\theta}(X_i)$ and corresponding indices $\widehat{Z}_i = X_i^\top \widehat{\theta}_i, i = 1, \ldots, n$, which are used as inputs for the subsequent LFR module. To obtain a global index direction, we aggregate the sample-level directions $\{\widehat{\theta}_i\}_{i=1}^n$ by their intrinsic mean on the unit sphere and denote the resulting estimator by $\widehat{\theta}$. While the DNN produces input-dependent directions $\widehat{\theta}(x)$, the single-index model assumes a common underlying direction $\theta_0$. Under this assumption, the sample-level directions can be viewed as noisy realizations of $\theta_0$, and their intrinsic mean provides a natural estimator of the global direction.

### 3.3. LFR Along the Learned Index

Given the estimated single-index values $\{\widehat{Z}_i\}_{i=1}^n$ produced by the DNN, we predict metric space-valued outputs using LFR along the one-dimensional index. Since the index values are estimated rather than directly observed, this step corresponds to an *errors-in-variables* version of LFR.

Specifically, for a query index value $\widehat{z} \in \mathbb{R}$ and a bandwidth parameter $h > 0$, the predicted output is obtained by solving

$$
\begin{aligned}
\widehat{\zeta}_h(\widehat{z}) &= \underset{y \in \Omega}{\arg\min}\, \widehat{M}_h(\widehat{z}, y), \\
\widehat{M}_h(\widehat{z}, y) &= n^{-1} \sum_{i=1}^n \widehat{w}_{in}(\widehat{z}, h)\, d^2(Y_i, y),
\end{aligned}
\tag{3}
$$

with weights $\widehat{w}_{in}(\widehat{z}, h) = \frac{1}{\widehat{\sigma}_0^2} K_h(\widehat{Z}_i - \widehat{z})[\widehat{\mu}_2 - \widehat{\mu}_1(\widehat{Z}_i - \widehat{z})]$, $\widehat{\mu}_j = n^{-1} \sum_{i=1}^n K_h(\widehat{Z}_i - \widehat{z})(\widehat{Z}_i - \widehat{z})^j$, $\widehat{\sigma}_0 = \widehat{\mu}_0 \widehat{\mu}_2 - \widehat{\mu}_1^2$.

Details on the numerical implementation of (3) for common metric spaces, including SPD matrices, networks, compositional data, and probability distributions, are provided in Appendix B.

### 3.4. Model Training

As loss function for training the model we use

$$\ell(\beta, h) = \frac{1}{n} \frac{\sum_{i=1}^{n} d^2(Y_i, \widehat{Y}_i)}{\widehat{V}_\oplus(Y)} + \frac{\lambda}{h}, \tag{4}$$

where $\widehat{Y}_i$ denotes the predicted output for $X_i$ and $\beta$ represents the parameter set of the DNN. Here $\widehat{V}_\oplus(Y) = n^{-1} \sum_{i=1}^{n} d^2(Y_i, \bar{Y}_\oplus)$ represents the sample Fréchet variance with $\bar{Y}_\oplus = \arg\min_{\omega \in \Omega} n^{-1} \sum_{i=1}^{n} d^2(Y_i, \omega)$ denoting the sample Fréchet mean. Dividing by $\widehat{V}_\oplus(Y)$ standardizes the squared prediction error, making the loss invariant to the overall scale of the output space. This normalization also stabilizes training and facilitates the selection of the tuning parameter $\lambda$ across different metric spaces.

The second term $\lambda/h$ penalizes excessively small bandwidth values in the LFR step, thereby controlling overfitting and encouraging appropriate smoothing along the learned single index. Importantly, the bandwidth $h$ is treated as a *learnable parameter* and is optimized jointly with the network parameters $\beta$, rather than being fixed or chosen in a separate tuning stage. This end-to-end estimation allows the degree of smoothing to adapt automatically to the data and avoids the need for manual bandwidth selection. The hyperparameter $\lambda > 0$ is selected via grid search by minimizing the cross-validated prediction error; see Appendix H for details.

The joint optimization of $(\beta, h)$ is performed using the iterative procedure summarized in Algorithm 1. At each training epoch, the current network parameters are used to compute single-index projections of the inputs, after which LFR is applied in a leave-one-out manner to obtain predicted outputs. The loss in (4) is then evaluated and used to update both $\beta$ and $h$, with early stopping based on validation performance.

## 4. Theoretical Properties

We establish approximation and rate guarantees for the proposed DeSI estimator. In particular, we show that sufficiently expressive neural networks can uniformly approximate the true single-index function, and we quantify how first-stage index estimation error propagates into the subsequent LFR step.

### 4.1. Uniform Approximation of the Index Direction

The approximation capability of the neural network for learning the single index is first established. We impose standard assumptions to ensure sufficient network capacity and identifiability of the index direction.

**Assumption 1** (Width and depth of neural network). The neural network $g(\cdot; \beta)$ has arbitrary finite depth and width (the number of neurons per layer can be chosen sufficiently large depending on the desired approximation error $\delta$).

---

**Algorithm 1** Deep Single-Index Fréchet Regression (DeSI)

**Require:** Training data $\{(X_i, Y_i)\}_{i=1}^{n}$, number of epochs $T$, learning rate $\eta$, penalty parameter $\lambda$.
**Ensure:** Estimated DNN parameters $\beta$ and bandwidth $h$.
1: Initialize DNN parameters $\beta^{(0)}$ and bandwidth $h^{(0)}$.
2: Split the data into a training set $\{(X_i, Y_i)\}_{i=1}^{k}$ and a validation set $\{(X_i, Y_i)\}_{i=k+1}^{n}$, with $k < n$.
3: Compute the sample Fréchet variance $\widehat{V}_\oplus(Y)$ using $\{Y_i\}_{i=1}^{k}$.
4: **for** $t = 1$ **to** $T$ **do**
5:     Compute single-index projections $\widehat{Z}_i^{(t)} = g(X_i; \beta^{(t)})$ for $i = 1, \ldots, k$.
6:     **for** $i = 1$ **to** $k$ **do**
7:         Using local Fréchet regression with bandwidth $h^{(t)}$, compute the predicted output $\widehat{Y}_i^{(t)}$ at index $\widehat{Z}_i^{(t)}$ based on the projected pairs $\{(\widehat{Z}_j^{(t)}, Y_j)\}_{j=1, j\neq i}^{k}$.
8:     **end for**
9:     Compute the training loss

$$\frac{1}{k} \frac{\sum_{i=1}^{k} d^2(Y_i, \widehat{Y}_i^{(t)})}{\widehat{V}_\oplus(Y)} + \frac{\lambda}{h^{(t)}}.$$

10:     Evaluate the validation loss using the validation set.
11:     Apply early stopping if the validation loss does not improve for five consecutive iterations.
12:     Update parameters $(\beta^{(t)}, h^{(t)})$ to $(\beta^{(t+1)}, h^{(t+1)})$ using gradient-based optimization with learning rate $\eta$.
13: **end for**

---

**Assumption 2** (Identifiability). For any $\theta \in \mathbb{S}^{p-1}$ with $\theta \neq \theta_0$, there exists $\delta > 0$ such that

$$\mathbb{P}\Big(d\big(\zeta(X^\top \theta_0), \zeta(X^\top \theta)\big) > \delta\Big) > 0.$$

**Assumption 3** (Density of the Index). The index $Z = g_0(X)$ admits a Lebesgue density bounded away from zero on a compact interval containing its essential support.

Assumption 1 guarantees sufficient expressive power of the neural network, while Assumptions 2 and 3 ensure identifiability of the true direction vector $\theta_0$.

**Theorem 4.1** (Uniform approximation of the estimated direction). *Suppose Assumptions 1–3 hold. For any $\delta > 0$, there exists a Leaky ReLU neural network $g(\cdot; \beta_\delta)$ such that*

$$\sup_{x \in \mathcal{D}} |g(x; \beta_\delta) - g_0(x)| < \delta,$$

$$\|\widehat{\theta} - \theta_0\|_2 < \delta/C_0,$$

*where $C_0$ denotes the radius of $\mathcal{D}$.*

Theorem 4.1 implies that the learned single index $\widehat{z} = g(X; \beta_\delta)$ uniformly approximates the true index $z = g_0(X)$.

Under the single-index structure, the corresponding aggregated estimator $\widehat{\theta}$ provides an approximation to the underlying direction $\theta_0$.

## 4.2. Convergence Rate of the DeSI Estimator

We next analyze the effect of index estimation error on the subsequent LFR step. Let $\widehat{\zeta}_h(\widehat{z})$ denote the LFR estimator evaluated at the estimated index using projected samples $\{(\widehat{Z}_i, Y_i)\}_{i=1}^n$ and let $\widetilde{\zeta}_h(z)$ denote the LFR estimator evaluated at the true index using the true projections $\{(Z_i, Y_i)\}_{i=1}^n$. To obtain convergence rates for the LFR step and the overall DeSI estimator, we require additional regularity conditions on the kernel, the data-generating process, and the geometry of the output space. These assumptions are standard in the literature on Fréchet regression with non-Euclidean outputs (Petersen & Müller, 2019; Chen & Müller, 2022; Schötz, 2022; Iao et al., 2025), and are stated and discussed in detail in Appendix C.

**Lemma 4.2.** *Suppose* $\|\widehat{z} - z\|_2 = O_p(b_n)$, $nh \to \infty$, *and* $h^{-3}b_n \to 0$. *Under Assumptions A1 and A4 stated in Appendix C,*

$$d\left\{\widehat{\zeta}_h(\widehat{z}), \widetilde{\zeta}_h(z)\right\} = O_p\left\{\left(h^{-2}b_n\right)^{1/(\psi_3-1)}\right\}.$$

Combining this result with the convergence rate of LFR evaluated at the true index yields the overall convergence rate of the DeSI estimator

$$\widehat{m}(\cdot) = \widehat{\zeta}_h(g(\cdot, \beta_\delta)).$$

**Theorem 4.3** (Convergence rate of DeSI). *For a new input $x$ independent of the training sample, let $\widehat{z} = g(x; \beta_\delta)$. Under Assumptions A1–A5 stated in Appendix C,*

$$d\big(\widehat{m}(x), m(x)\big) = O_p\Big\{\left(h^{-2}b_n\right)^{1/(\gamma_3-1)}\Big\} + $$
$$O_p\Big\{h^{2/(\gamma_1-1)} + (nh)^{-1/(2\gamma_2-2)}\Big\}.$$

The first term captures the uncertainty caused by learning of single indexes through DNN and the induced error in the subsequent analysis, and the last two terms reflect the convergence of LFR. For the metric spaces discussed in example 1 to 3, $\gamma_1 = \gamma_2 = \gamma_3 = 2$ and the convergence rate reduces to $O_p(h^{-2}b_n + h^2 + (nh)^{-1/2})$.

## 5. Simulations

We conduct simulation studies under three settings corresponding to distinct metric spaces introduced in Examples 1–3: the space of SPD matrices equipped with the log-Cholesky metric, the space of networks equipped with the Frobenius metric, and the space of univariate probability distributions equipped with the Wasserstein metric. Each

setting is tested across sample sizes $n = 200, 500, 1000$, with 200 Monte Carlo replications per scenario. Across these settings, we compare the proposed DeSI method with existing approaches for Fréchet regression, including global Fréchet regression (GFR) (Petersen & Müller, 2019), single-index Fréchet regression (IFR) (Bhattacharjee & Müller, 2023), and deep Fréchet regression (DFR) (Iao et al., 2025). DFR does not currently have an implementation for SPD matrix-valued outputs under the log-Cholesky metric and is therefore not included in the SPD setting.

For the $q$-th Monte Carlo replication, let $\widehat{m}_q$ denote the estimated regression function and $m$ the true regression function. The mean prediction error is computed using 100 independent test samples as

$$\text{MPE}_q = 100^{-1} \sum_{i=1}^{100} d\{\widehat{m}_q(X_i^{\text{test}}), m(X_i^{\text{test}})\}$$

where $d(\cdot, \cdot)$ is the corresponding metric on the output space.

Both IFR and DeSI provide estimators of the true index direction $\theta_0$. While IFR produces a single global estimate $\widehat{\theta}$, DeSI yields subject-specific estimates $\widehat{\theta}_i$, $i = 1, \ldots, n$. Since $\theta_0$ lies on the unit sphere, we aggregate the DeSI estimates by computing the intrinsic sample mean of $\{\widehat{\theta}_i\}_{i=1}^n$ on the sphere, which is then taken as $\widehat{\theta}$. The accuracy of index estimation is quantified by the $L_2$ distance between $\theta_0$ and $\widehat{\theta}$.

### 5.1. SPD Matrices

We consider a regression setting in which the output is an SPD matrix equipped with the log-Cholesky metric defined in Example 1 and the inputs are Euclidean vectors. For each input vector $X_i \in \mathbb{R}^4$, $i = 1, \ldots, n$, we generate the corresponding SPD matrix as follows.

**Input.** The inputs are $X_i = (X_{i1}, X_{i2}, X_{i3}, X_{i4})^\top$, $i = 1, \ldots, n$, where $X_{i1} \sim \text{Unif}(1, 2)$, $X_{i2}, X_{i3} \sim \text{Unif}(0, 1)$, $X_{i4} \sim \text{Unif}(-1, 0)$ independently across coordinates and subjects. We fix

$$\theta_{\text{raw}} = (0.1, 0.5, 0, -0.1)^\top, \qquad \theta_0 = \frac{\theta_{\text{raw}}}{\|\theta_{\text{raw}}\|}, \quad (5)$$

and define the single index $Z_i$ for subject $i$ as $Z_i = \theta_0^\top X_i$.

**Output.** Let $q = 3$ be the matrix dimension. We construct a fixed orthonormal matrix $Q \in \mathbb{R}^{q \times q}$ using a discrete Legendre-type basis, which is shared across all simulated samples. For each $i$, the eigenvalues are deterministic functions of the single index $Z_i$, $\lambda_{i1} = Z_i$, $\lambda_{i2} = Z_i^2$, $\lambda_{i3} = e^{Z_i}$, and are assembled into the diagonal matrix $\Lambda(Z_i) = \text{diag}(\lambda_{i1}, \lambda_{i2}, \lambda_{i3})$. The SPD matrices are then

$$Y_i = Q\Lambda(Z_i)Q^\top + E_i, \ i = 1, \ldots, n,$$

where $E_i = \text{diag}(\varepsilon_{i1}, \varepsilon_{i2}, \varepsilon_{i3})$ and $\varepsilon_{i1}, \varepsilon_{i2}, \varepsilon_{i3} \overset{\text{i.i.d.}}{\sim}$ $\text{Unif}(-10^{-3}, 10^{-3})$.

## 5.2. Networks

We consider a network-valued regression setting in which the output is a weighted network, represented by its graph Laplacian and equipped with the Frobenius metric.

**Input:** $\theta_0$ is the same as (5) and the input vectors are $X_i = (X_{i1}, X_{i2}, X_{i3}, X_{i4})^\top, i = 1, \ldots, n$, where $X_{ij} \overset{\text{i.i.d.}}{\sim}$ $\text{Unif}(0, 1)$, and the single index $Z_i$ for subject $i$ is defined as $Z_i = \theta_0^\top X_i$.

**Output:** We first generate a symmetric binary adjacency matrix $\mathbf{A} \in \{0, 1\}^{q \times q}$ with $q = 10$, where for $k < \ell$, $A_{k\ell} = A_{\ell k} \overset{\text{i.i.d.}}{\sim} \text{Bernoulli}(0.3)$ and $A_{kk} = 0$. Given $\mathbf{A}$, we define a symmetric weighted adjacency matrix $\mathbf{E}_i = (E_{i,k\ell})$ by

$$E_{i,k\ell} = \begin{cases} \sin\left(\dfrac{(k+\ell)\pi}{2q}\right) \dfrac{2 + Z_i^2}{|Z_i| + 1} + \varepsilon_{i,k\ell}, & k < \ell, \\ E_{i,\ell k}, & k > \ell, \\ 0, & k = \ell, \end{cases}$$

where $\varepsilon_{i,k\ell} \sim \text{Unif}(-0.02, 0.02)$ are independent noise terms and $E_{i,k\ell} = 0$ when $A_{k,\ell} = 0$. Let $D_i$ denote the degree matrix associated with $\mathbf{E}_i$, defined as $D_i = \text{diag}(\mathbf{E}_i \mathbf{1}_q)$. The network response is represented by its graph Laplacian $Y_i = D_i - \mathbf{E}_i$.

## 5.3. Univariate Probability Distributions

**Input:** The true index direction $\theta_0$ is set as in (5). The predictors are $X_i = (X_{i1}, X_{i2}, X_{i3}, X_{i4})^\top$, $i = 1, \ldots, n$. For each $i$, we first generate a Gaussian vector $U_i = (U_{i1}, U_{i2}, U_{i3}, U_{i4})^\top \sim \mathcal{N}(0, \Sigma)$, where $\Sigma_{jj} = 1$ and $\Sigma_{jk} = \rho$ for $j \neq k$, with $\rho = 0.25$. Each component is then transformed as $X_{ij} = 2\Phi(U_{ij}) - 1$, where $\Phi(\cdot)$ denotes the standard normal cumulative distribution function.

**Output:** We consider three transformation functions for the single index $Z_i$:

$$\psi(Z_i) = \begin{cases} Z_i, & \text{(linear)}, \\ Z_i^2, & \text{(quadratic)}, \\ \exp(Z_i), & \text{(exponential)}. \end{cases}$$

Conditional on $Z_i$, the response distribution is generated as a Gaussian $\mathcal{N}(\mu_i, \sigma_i^2)$, where

$$\mu_i = \psi(Z_i) + \xi_i, \quad \xi_i \sim \mathcal{N}(0, 0.25^2),$$

$$\sigma_i \sim \text{Exp}\left(\frac{1}{\eta_i}\right), \quad \eta_i = \frac{\exp(Z_i)}{1 + \exp(Z_i)}.$$

Each distributional output is represented by its quantile function on a fixed grid, with the true quantile function given by $Q_i(p) = \mu_i + \sigma_i \Phi^{-1}(p)$ for $p \in (0, 1)$, corresponding to a shift-scale family.

The MPE, averaged over 200 Monte Carlo replications along with the corresponding standard deviations, is reported in Table 1 for all three simulation settings. Across different metric spaces and sample sizes, DeSI consistently attains the lowest or among the lowest prediction errors. Moreover, its performance advantage becomes increasingly pronounced as the sample size grows.

Table 2 summarizes the accuracy of estimating the single-index direction. Across all metric spaces and experimental settings, DeSI yields uniformly smaller estimation error than IFR, with more pronounced improvements in the network and nonlinear distributional settings. These results demonstrate the effectiveness of DeSI in both predictive performance and recovery of the underlying index structure. Additional boxplots stratified by sample size are provided in Appendix E.

## 6. Data Application

We illustrate the proposed Deep Single-Index Fréchet Regression (DeSI) method using compositional mood data from the Survey of Unemployed Workers in New Jersey (Krueger et al., 2011), collected between 2009 and 2010. The study is based on a stratified random sample of unemployed individuals at the time of enrollment. After excluding incomplete records, the analysis includes $n = 2284$ participants with fully observed measurements.

The primary outcome of interest is the distribution of time spent at home across four affective states: bad, low or irritable, mildly pleasant, and very good. For each individual, the mood outcome is recorded as a compositional vector $W = (W_1, W_2, W_3, W_4)^\top$, where $W_j$ denotes the proportion of time spent in the $j$th mood category. As discussed in Example 4, we apply a component-wise square-root transformation,

$$Y = (\sqrt{W_1}, \sqrt{W_2}, \sqrt{W_3}, \sqrt{W_4}),$$

which maps the compositional data to the positive orthant of the unit sphere $\mathbb{S}^3$ equipped with the geodesic metric $d_g$.

The predictors consist of a baseline covariate vector $X = (X_1, \ldots, X_{10})$, constructed from questionnaire data and capturing social, demographic, economic, and labor market characteristics. These covariates include self-reported life satisfaction, educational attainment, marital status, number of children, household size, total household income, job separation type, duration of job search, and measures of financial resources such as savings and credit card debt.

We apply DeSI, global Fréchet regression (GFR) (Petersen &

*Table 1.* Mean prediction error, reported as mean (standard deviation) over 200 Monte Carlo replications, across different metric spaces and sample sizes. Methods compared include deep single-index Fréchet regression (DeSI; proposed), global Fréchet regression (GFR; Petersen & Müller, 2019), single-index Fréchet regression (IFR; Bhattacharjee & Müller, 2023), and deep Fréchet regression (DFR; Iao et al., 2025). Boldface indicates the best performance within each setting.

| Method | $n$ | SPD | Networks | Dist. (Lin.) | Dist. (Quad.) | Dist. (Exp.) |
|---|---|---|---|---|---|---|
| DeSI | 200 | **0.0095 (0.0101)** | 0.2214 (0.2838) | 0.1582 (0.0213) | **0.2031 (0.0668)** | **0.1377 (0.0131)** |
| | 500 | **0.0067 (0.0078)** | **0.0790 (0.1274)** | **0.0461 (0.0143)** | **0.1570 (0.0564)** | **0.0985 (0.0203)** |
| | 1000 | **0.0057 (0.0054)** | **0.0531 (0.0428)** | **0.0344 (0.0121)** | **0.0892 (0.0431)** | **0.0690 (0.0115)** |
| DFR | 200 | — | **0.1389 (0.0317)** | 0.2441 (0.0353) | 0.2420 (0.0320) | 0.2455 (0.0335) |
| | 500 | — | 0.0840 (0.0176) | 0.1857 (0.0188) | 0.1890 (0.0174) | 0.1868 (0.0193) |
| | 1000 | — | 0.0591 (0.0107) | 0.1461 (0.0127) | 0.1475 (0.0118) | 0.1469 (0.0127) |
| GFR | 200 | 0.0153 (0.0017) | 0.5705 (0.1061) | **0.0744 (0.0195)** | 0.2662 (0.0191) | 0.1623 (0.0165) |
| | 500 | 0.0153 (0.0015) | 0.5696 (0.1021) | 0.0486 (0.0129) | 0.2534 (0.0163) | 0.1428 (0.0118) |
| | 1000 | 0.0153 (0.0014) | 0.5663 (0.0977) | 0.0351 (0.0094) | 0.2498 (0.0157) | 0.1418 (0.0095) |
| IFR | 200 | 0.0128 (0.0045) | 0.5598 (0.1821) | 0.2200 (0.0880) | 0.2628 (0.0430) | 0.2705 (0.1059) |
| | 500 | 0.0127 (0.0046) | 0.5769 (0.2144) | 0.2053 (0.0805) | 0.2605 (0.0369) | 0.2605 (0.1120) |
| | 1000 | 0.0101 (0.0038) | 0.4927 (0.1825) | 0.1644 (0.0571) | 0.2272 (0.0612) | 0.1890 (0.0676) |

*Table 2.* Prediction error for estimating the single-index direction $\theta_0$, reported as mean (standard deviation) over 200 Monte Carlo replications. Results compare deep single-index Fréchet regression (DeSI) with single-index Fréchet regression (IFR; Bhattacharjee & Müller, 2023) across different metric spaces and sample sizes. Boldface indicates the smaller estimation error.

| Method | $n$ | SPD | Networks | Dist. (Lin.) | Dist. (Quad.) | Dist. (Exp.) |
|---|---|---|---|---|---|---|
| DeSI | 200 | **0.2170 (0.2640)** | **0.1387 (0.3582)** | **0.0213 (0.0001)** | **0.0262 (0.0020)** | **0.0213 (0.0001)** |
| | 500 | **0.1210 (0.2631)** | **0.0297 (0.1722)** | **0.0208 (0.0000)** | **0.0255 (0.0028)** | **0.0207 (0.0000)** |
| | 1000 | **0.0640 (0.0181)** | **0.0023 (0.0143)** | **0.0206 (0.0000)** | **0.0220 (0.0014)** | **0.0164 (0.0000)** |
| IFR | 200 | 0.3101 (0.1612) | 0.2353 (0.2171) | 0.0783 (0.0058) | 0.2767 (0.0113) | 0.1037 (0.0083) |
| | 500 | 0.3222 (0.1663) | 0.2882 (0.3145) | 0.0703 (0.0053) | 0.2740 (0.0104) | 0.1101 (0.0093) |
| | 1000 | 0.2150 (0.1001) | 0.1744 (0.1683) | 0.0451 (0.0015) | 0.2211 (0.0123) | 0.0474 (0.0024) |

Müller, 2019), single-index Fréchet regression (IFR) (Bhattacharjee & Müller, 2023) to the data. DFR is is not included, as the current implementation does not support compositional outcomes under the geodesic metric; extending DFR to this setting would require metric-specific modifications. Model performance is assessed using 100 Monte Carlo replications, each consisting of a 10-fold cross-validation procedure. Table 3 reports the MPE, averaged over the Monte Carlo replications, together with the corresponding standard deviations for each method. DeSI achieves the lowest prediction error among the three methods, yielding an MPE reduction of approximately 6% relative to IFR and 34% relative to GFR. These results indicate that the proposed deep single-index structure effectively captures nonlinear covariate effects while retaining the local adaptivity of Fréchet regression.

Beyond predictive performance, DeSI also provides an interpretable summary of the index direction. The aggregated direction estimator is

$$\widehat{\theta} = (0.9915,\ 0.0297,\ -0.039,\ -0.023,\ 0.0374,$$
$$-0.0309,\ 0.0316,\ 0.007,\ 0.094,\ 0.0427)^{\top}.$$

The first component, associated with self-reported life sat-

*Table 3.* MPE for the New Jersey compositional mood data application. Results are reported as mean (standard deviation) over 100 Monte Carlo replications, comparing deep single-index Fréchet regression (DeSI), single-index Fréchet regression (IFR; Bhattacharjee & Müller, 2023), and global Fréchet regression (GFR; Petersen & Müller, 2019).

| | DeSI | IFR | GFR |
|---|---|---|---|
| MPE | **0.4658 (0.0012)** | 0.4947 (0.0101) | 0.6229 (0.0044) |

isfaction, is by far the largest in magnitude. This indicates that life satisfaction is the primary driver of variation in mood composition. In contrast, the remaining variables have much smaller coefficients, suggesting they play more minor roles in comparison. To further illustrate the implications of the learned index, Figure 3 shows the distribution of mood compositions across increasing levels of life satisfaction. A higher level of life satisfaction is associated with a marked decrease in negative moods, particularly bad mood and with a corresponding increase in mildly pleasant and very good moods. This pattern aligns closely with the dominant index component identified by DeSI and provides empirical support for both the predictive effectiveness and interpretability of the proposed approach.

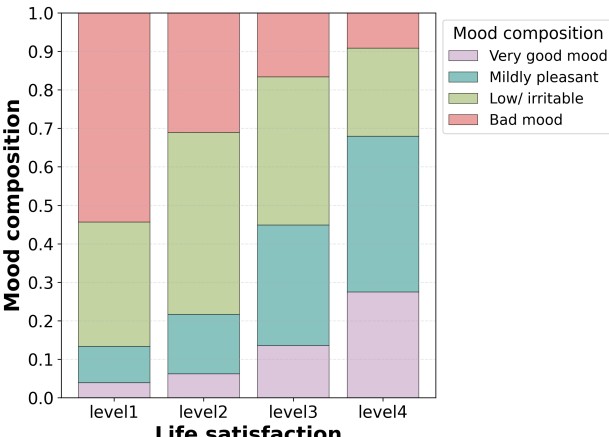

*Figure 3.* Mood composition across different life satisfaction levels.

## 7. Discussion

The proposed framework provides interpretability through a single-index structure. By reducing the dependence of the regression function on high-dimensional predictors to a one-dimensional projection, the model yields a direction whose components quantify the relative importance of predictors. In contrast to standard DNNs, which typically operate as black-box models, the proposed approach combines the flexibility of deep learning with the interpretability of single-index models. This is particularly valuable in regression scenarios with metric space-valued outputs, where standard linear interpretations are not available due to the lack of algebraic structure in the response space.

Although the proposed model is based on the single-index assumption, it remains effective beyond this setting. In scenarios where the true relationship does not strictly follow a single-index structure, the learned projection can still serve as a meaningful low-dimensional representation of the predictors when aiming at the prediction of random objects. This suggests that the method is anticipated to be robust to a certain level of model misspecification, which is indeed supported by the empirical results in Appendix F. In such cases, the estimated index should be interpreted as an optimal predictive projection rather than a recovery of a true underlying direction.

The single-index framework can be extended to settings where a single index alone is insufficient to capture the full complexity of the relationship between predictors and the response. A natural extension is a multi-index model, in which multiple projections are learned and local Fréchet regression is performed in a higher-dimensional index space. While this would increase modeling flexibility and potentially improve predictive performance, it also reduces interpretability, as multiple directions must be interpreted jointly. Under-

standing how to balance flexibility and interpretability in such extensions is an important direction for future work.

## Impact Statement

This work advances machine learning and statistics by developing regression methodology for complex, non-Euclidean outputs. The proposed framework is general-purpose and applicable to scientific and engineering problems involving structured data such as distributions and networks. We do not anticipate any direct negative societal or ethical impacts; as with other general modeling tools, downstream effects depend on the application context, and responsible use in accordance with domain-specific ethical standards is encouraged.

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

# A. Fréchet Mean, Conditional Fréchet Mean and Local Fréchet Regression

## A.1. Fréchet Mean and Conditional Fréchet Mean

In classical statistics, the population mean can be defined as the optimal point estimator under a squared error loss. Specifically, let $Y$ be a real-valued random variable in $\mathbb{R}$. The expectation of $Y$ is uniquely characterized as the minimizer of the expected squared deviation:

$$\mathbb{E}[Y] = \arg \min_{y \in \mathbb{R}} \mathbb{E}\left[(Y - y)^2\right].$$

This identity is derived by expanding the objective function, $\mathbb{E}[(Y - y)^2] = \mathbb{E}[Y^2] - 2y\mathbb{E}[Y] + y^2$, and observing that the first-order optimality condition

$$\frac{d}{dy}\mathbb{E}[(Y - y)^2] = -2\mathbb{E}[Y] + 2y = 0$$

yields $y = \mathbb{E}[Y]$. For a random pair $(X, Y) \in \mathbb{R}^p \times \mathbb{R}$, the conditional expectation $\mathbb{E}[Y \mid X = x]$ admits a parallel characterization:

$$\mathbb{E}[Y \mid X = x] = \arg \min_{y \in \mathbb{R}} \mathbb{E}\left[(Y - y)^2 \mid X = x\right].$$

The mean and conditional mean could be naturally extended to random objects taking values in a general metric space $(\Omega, d)$. For a random variable $Y$ taking values in $\Omega$, the Fréchet mean is defined as the minimizer of the expected squared distance (Fréchet, 1948):

$$\mathbb{E}_\oplus[Y] = \arg \min_{y \in \Omega} \mathbb{E}\left[d^2(y, Y)\right].$$

Similarly, for a random pair $(X, Y) \in \mathbb{R}^p \times \Omega$, the conditional Fréchet mean is given by:

$$\mathbb{E}_\oplus[Y \mid X = x] = \arg \min_{y \in \Omega} \mathbb{E}\left[d^2(y, Y) \mid X = x\right].$$

These definitions simplify to the classical mean and conditional means when $\Omega = \mathbb{R}$ and $d(y, Y) = |y - Y|$.

## A.2. Local Fréchet Regression

For a scalar output $Y \in \mathbb{R}$ and an input $Z \in \mathbb{R}$, we consider the estimation of the conditional mean function $m(z) = \mathbb{E}[Y \mid Z = z]$. Local linear regression approximates $m(z)$ by solving a kernel-weighted least squares problem, which provides a first-order Taylor approximation of the regression function in the neighborhood of $z$ (Fan, 2018):

$$(\beta_0, \beta_1) = \arg \min_{\beta_0, \beta_1 \in \mathbb{R}} \mathbb{E}\left[K_h(Z - z)\left(Y - \beta_0 - \beta_1(Z - z)\right)^2\right],$$

where $K_h(\cdot) = h^{-1}K(\cdot/h)$ denotes a kernel function with bandwidth $h > 0$.

The coefficients $(\beta_0, \beta_1)$ admit closed-form expressions determined by the weighted moments of the covariate and output. Let $\mu_k = \mathbb{E}[K_h(Z - z)(Z - z)^k]$ and $\xi_k = \mathbb{E}[K_h(Z - z)(Z - z)^k Y]$. The solution for the intercept $\beta_0$, which serves as our estimator $\widetilde{m}(z)$, is given by:

$$\widetilde{m}(z) = \beta_0 = \frac{\mu_2 \xi_0 - \mu_1 \xi_1}{\mu_0 \mu_2 - \mu_1^2}.$$

By linearity of the expectation, the local linear regression estimator can be represented as a weighted mean of the random output: $\widetilde{m}(z) = \mathbb{E}[w(z; Z, h)Y]$, where the equivalent kernel weight function is defined as

$$w(z; Z, h) = \frac{K_h(Z - z)\left(\mu_2 - \mu_1(Z - z)\right)}{\mu_0 \mu_2 - \mu_1^2}. \tag{6}$$

Analogous to the characterization of the mean in Appendix A, the local linear estimator $\widetilde{m}(z)$ can be interpreted as a weighted mean in Euclidean space. Specifically, it solves the following localized risk minimization problem:

$$\widetilde{m}(z) = \arg \min_{y \in \mathbb{R}} \mathbb{E}\left[w(z; Z, h)(Y - y)^2\right].$$

This formulation highlights that local linear regression is effectively a local weighted mean problem, where the weight function $w(z; Z, h)$ adaptively accounts for the local geometry and density of the covariate distribution.

The variational perspective above extends naturally to non-Euclidean settings. Suppose the output $Y$ resides in a general metric space $(\Omega, d)$, while the input $Z \in \mathbb{R}$ remains a scalar. The local Fréchet regression function $\widetilde{m}(z)$ is defined as:

$$\widetilde{m}(z) = \arg\min_{y \in \Omega} \mathbb{E}\Big[w(z; Z, h)d^2(Y, y)\Big], \tag{7}$$

where the weight function $w$ is defined as in Equation (6). This characterization demonstrates that local Fréchet regression generalizes local linear regression by replacing the Euclidean output $Y \in \mathbb{R}$ and the squared error loss with a metric space-valued output $Y \in \Omega$ and the squared metric distance $d^2(Y, y)$.

## B. Optimization Details

### B.1. SPD Matrices

For symmetric positive-definite (SPD) matrices, we employ the log-Cholesky metric in the local Fréchet regression step. Let $Y_i \in \Omega$ admit the Cholesky decomposition $Y_i = L_i L_i^\top$, where $L_i$ is lower triangular with positive diagonal entries. Define the log-Cholesky mapping $F_{LC} : \Omega \to \mathbb{R}^{q(q+1)/2}$ by

$$F_{LC}(Y_i) = \big(\ell_{jk}(Y_i)\big)_{1 \le j, k \le q}, \qquad \ell_{jk}(Y_i) = \begin{cases} L_{jk}, & \text{if } j > k, \\ \log L_{jj}, & \text{if } j = k, \\ 0, & \text{if } j < k. \end{cases}$$

This map $F_{LC}$ is a global isometry from $(\Omega, d_{\mathrm{LC}})$ onto its image in the Euclidean space $\mathbb{R}^{q(q+1)/2}$ equipped with the Frobenius norm $\|\cdot\|_F$, i.e.,

$$d_{\mathrm{LC}}(Y_1, Y_2) = \big\|F_{LC}(Y_1) - F_{LC}(Y_2)\big\|_F \quad \forall Y_1, Y_2 \in \Omega.$$

Given weights $w_i$ with $\sum_{i=1}^n w_i > 0$, the weighted local Fréchet mean is defined as

$$\widehat{Y} = \arg\min_{y \in \Omega} \frac{1}{n} \sum_{i=1}^n w_i \, d_{\mathrm{LC}}^2(Y_i, y)$$

$$= \arg\min_{y \in \Omega} \frac{1}{n} \sum_{i=1}^n w_i \big\|F_{LC}(Y_i) - F_{LC}(y)\big\|_F^2.$$

Since $F_{LC}$ is an isometry, minimizing the Fréchet objective under $d_{\mathrm{LC}}$ is equivalent to minimizing the weighted squared Euclidean distance in the vector space $\mathbb{R}^{q(q+1)/2}$. Let

$$F_{LC}^\mu := \Big(\sum_{i=1}^n w_i\Big)^{-1} \sum_{i=1}^n w_i \, F_{LC}(Y_i)$$

be the weighted Euclidean mean of the vectors $F_{LC}(Y_i)$. Then the unique minimizer of the quadratic objective

$$y \mapsto \sum_{i=1}^n w_i \big\|F_{LC}(Y_i) - F_{LC}(y)\big\|_F^2$$

is attained at the point $y^*$ satisfying $F_{LC}(y^*) = F_{LC}^\mu$. Because $F_{LC}$ is bijective (every strictly lower-triangular matrix with positive diagonal entries corresponds to a unique SPD matrix via exponentiation on the diagonal and reconstruction), there exists a unique $y^* \in \Omega$ such that

$$F_{LC}(y^*) = F_{LC}^\mu.$$

Explicitly, if we write $F_{LC}^\mu = (\bar{\ell}_{jk})_{1 \le j, k \le q}$ with $\bar{\ell}_{jk} = 0$ for $j < k$, where the average is the element-wise mean for $n$ samples, then the corresponding Cholesky factor $L^*$ is given by

$$L_{jk}^* = \begin{cases} \bar{\ell}_{jk}, & j > k, \\ \exp(\bar{\ell}_{jj}), & j = k, \\ 0, & j < k, \end{cases}$$

and the unique Fréchet mean is

$$\widehat{Y} = L^*(L^*)^\top.$$

Thus the weighted Fréchet mean under the log-Cholesky metric exists, is unique, and admits the closed-form expression above.

## B.2. Networks

For networks, we represent each network by its graph Laplacian matrix. Let $\Omega$ denote the set of valid graph Laplacian matrices (i.e., symmetric positive semi-definite matrices with nonpositive off-diagonal entries and row sums equal to zero) corresponding to the networks under consideration. For each observed network $Y_i \in \Omega$, let $L_i$ be its associated graph Laplacian matrix. We equip $\Omega$ with the Frobenius metric

$$d_F(Y_1, Y_2) = \|Y_1 - Y_2\|_F = \Big( \sum_{j,k=1}^{q} \big((Y_1)_{jk} - (Y_2)_{jk}\big)^2 \Big)^{1/2}.$$

It is well known that the map from a network (weighted adjacency matrix or unweighted adjacency structure) to its graph Laplacian is one-to-one (Zhou & Müller, 2022). In other words, different networks produce distinct graph Laplacians, and every valid graph Laplacian corresponds to exactly one underlying network.

Given weights $w_i$ with $\sum_{i=1}^{n} w_i > 0$, the weighted local Fréchet mean at a query point $t^*$ is defined as

$$\widehat{Y}(t^*) = \underset{y \in \Omega}{\arg\min} \frac{1}{n} \sum_{i=1}^{n} w_i \, d_F^2(Y_i, y)$$

$$= \underset{y \in \Omega}{\arg\min} \frac{1}{n} \sum_{i=1}^{n} w_i \, \|L_i - y\|_F^2,$$

where $L_i$ is the graph Laplacian of the $i$-th observed network $Y_i$. Although the objective

$$y \mapsto \sum_{i=1}^{n} w_i \|L_i - y\|_F^2$$

has the same algebraic form as in the SPD matrix setting, the analogous weighted average

$$\tilde{y} = \left( \sum_{i=1}^{n} w_i \right)^{-1} \sum_{i=1}^{n} w_i L_i$$

does not necessarily belong to the graph Laplacian space $\Omega$, since the weights $w_i$ may be negative. We therefore approximate the optimizer by projecting $\tilde{y}$ onto $\Omega$. Specifically, we define

$$\hat{y} = \Pi_\Omega(\tilde{y}) = \underset{y \in \Omega}{\arg\min} \|\tilde{y} - y\|_F^2.$$

The resulting matrix $\hat{y}$ is used as the optimizer for the object function.

## B.3. Compositional Data

For compositional data, as the optimization process does not have a closed form solution, we applied the gradient descent method on sphere with a fixed step-size (Amari, 1998). This algorithm is differentiable with respect to the parameters of the DNN and the minimizer is also unique (Petersen & Müller, 2019).

## B.4. Univariate Probability Distributions

For univariate probability distributions, we adopt Wasserstein regression that can accurately predict distributional outputs even when the sample sizes differ across distributions, including cases where some distributions have very small sample sizes while others have large ones (Zhou & Müller, 2024). Wasserstein regression solves a constrained convex quadratic

optimization problem with linear constraints using empirical measures and returns a discretized version of the unique predicted quantile function. Such constrained convex quadratic optimization problem is mathematically expressed as

$$\underset{x}{\text{minimize}} \quad \frac{1}{2}x^{\top}Hx + c^{\top}x$$
$$\text{subject to} \quad Ax = b_0,$$
$$Gx \leq b_1,$$

where the matrix $H$ is positive semi-definite, thereby guaranteeing that the objective function is convex and that the problem belongs to the class of convex optimization problems.

The optimization step is implemented by the `CVXPYlayers` to embed differentiable optimization problems as layers within deep learning architectures (Agrawal et al., 2019).

## C. Assumptions for Local Fréchet Regression

We make the following assumptions to establish the asymptotic convergence rate of the proposed estimator.

**Assumption A1** (Kernel conditions). The univariate kernel function $K(\cdot)$ satisfies

$$\int_{\mathbb{R}} K(u)\, u^4\, du < \infty \text{ and } \int_{\mathbb{R}} K^2(u)\, u^6\, du < \infty.$$

Moreover, $K(\cdot)$ is Lipschitz continuous and has a compact support.

**Assumption A2** (Smoothness of densities). The marginal density $f_Z(\cdot)$ of $Z$ and the conditional density $f_{Z|Y}(\cdot, y)$ of $Z$ given $Y = y$ exist and are twice continuously differentiable, with the latter holding uniformly for all $y \in \Omega$. Moreover,

$$\sup_{z,y} \left| \frac{\partial^2}{\partial z^2} f_{Z|Y}(z,y) \right| < \infty.$$

In addition, for any open set $A \subset \Omega$, the mapping

$$z \ \mapsto \ \int_A dF_{Y|Z}(z,y)$$

is continuous.

**Assumption A3** (Existence, uniqueness, and separation). The minimizers $\zeta(z)$, $\widetilde{\zeta}(z)$, $\widetilde{\zeta}(\widehat{z})$, and $\widehat{\zeta}(\widehat{z})$ of the local Fréchet regression criteria exist and are unique.

$$\inf_{d\{\zeta(z),y\}>\epsilon} \left[ M(z,y) - M(z,\zeta(z)) \right] > 0,$$
$$\liminf_{h\to 0} \inf_{d\{\zeta(z),y\}>\epsilon} \left[ M_h(z,y) - M_h\{z,\zeta(z)\} \right] > 0,$$
$$P\left( \inf_{d\{\widetilde{\zeta}_h(z),y\}>\epsilon} \left[ \widetilde{M}_h(y,z) - \widetilde{M}_h\{\widetilde{\zeta}_h(z),y\} \right] > 0 \right) \to 1,$$
$$P\left( \inf_{d\{\widehat{\zeta}_h(z),y\}>\epsilon} \left[ \widehat{M}_h(z,y) - \widehat{M}_h\{\widehat{\zeta}_h(z),y\} \right] > 0 \right) \to 1.$$

**Assumption A4** (Local curvature conditions). There exist constants $\eta_1, \eta_2, \eta_3 > 0$, $C_1, C_2, C_3 > 0$, and $\gamma_1, \gamma_2, \gamma_3 > 1$ such that

$$\inf_{d\{\zeta(z),y\}<\eta_1} \left[ M(z,y) - M\{z,\zeta(z)\} - C_1 d\{\zeta(z),y\}^{\gamma_1} \right] \geq 0,$$
$$\liminf_{h\to 0} \inf_{d\{\widetilde{\zeta}_h(z),y\}<\eta_2} \left[ M_h(z,y) - M_h\{z,\widetilde{\zeta}_h(z)\} - C_2 d\{\widetilde{\zeta}_h(z),y\}^{\gamma_2} \right] \geq 0,$$
$$P\left( \inf_{d\{\widehat{\zeta}_h(z),y\}<\eta_3} \left[ \widetilde{M}_h(z,y) - \widetilde{M}_h\{z,\widehat{\zeta}_h(z)\} - C_3\, d\{\widehat{\zeta}_h(z),y\}^{\gamma_3} \right] \geq 0 \right) \ \to \ 1.$$

**Assumption A5** (Entropy condition). Let $B_\phi(y) \subset \Omega$ denote the ball of radius $\phi$ centered at $y$, and let $N\{\epsilon, B_\phi(y), d\}$ be its covering number with respect to balls of radius $\epsilon$ under metric $d$. Then for any $y \in \Omega$,

$$\int_0^1 \left[1 + \log N\{\phi\epsilon, B_\phi(y), d\}\right]^{1/2} d\epsilon = O(1) \text{ as } \phi \to 0.$$

Assumptions A1 to A4 are required to guarantee the distance between $\widehat{\zeta}_h(\widehat{z})$ and $\widetilde{\zeta}_h(z)$ not too large (Iao et al., 2025). Assumption A1 is the common condition on kernel in local Fréchet regression. Assumptions A2 to A4 are required to guarantee the distance between $\widehat{\zeta}_h(\widehat{z})$ and $\widetilde{\zeta}_h(z)$ not too large (Iao et al., 2025), which impose standard regularity conditions to ensure the well-posedness and asymptotic validity of local Fréchet regression. Assumption A2 requires smoothness and boundedness of the joint and conditional densities of the projected covariate and output, which guarantees regular local behavior and justifies kernel-based approximations. Assumption A3 ensures existence, uniqueness, and uniform separation of the Fréchet regression minimizers for both population-level and sample-based objective functions, providing identifiability and consistency. Assumption A4 imposes local curvature conditions around the Fréchet regression targets, ensuring sufficient growth of the objective function away from its minimizer and allowing for rate derivations and asymptotic expansions. Assumption A5 needs to be verified on a case-by-case basis and for distributional responses requires a space of finite dimensionality, such as a shift-scale family. For distributional responses, this assumption can be bypassed by pertaining to quantile function representations, which form a convex subset of the Hilbert space $L^2$ and applying results of Petersen et al. (2019).

## D. Proofs

In the following theorem, we first prove that minimizing training loss is equivalent to minimizing the distance between $g(X; \beta)$ and $g_0(X)$. Then our training procedure is actually minimizing distance between index functions $g$ and $g_0$ and in the meantime, minimizing the distance between $\widehat{\theta}$ and $\theta_0$. According to Theorem 1 of Yarotsky (2017), as long as the activation function is continuous and piecewise linear and with an affine output layer, deep neural networks achieve essentially the same approximation guarantees as standard ReLU networks, up to universal constants. The universal approximation of the single index parameter $\theta_0$ is established as follows.

*Proof of Theorem 4.1.* We first show that the activation function and $g_0$ have the following properties:

- Continuity: At the breakpoint $u = 0$, we have

$$\lim_{u \to 0^-} \sigma(u) = \lim_{u \to 0^-} \alpha u = 0, \qquad \lim_{u \to 0^+} \sigma(u) = \lim_{u \to 0^+} u = 0, \qquad \sigma(0) = 0.$$

  Thus $\sigma$ is continuous at $u = 0$. On the open intervals $(-\infty, 0)$ and $(0, \infty)$ it is continuous. Hence $\sigma$ is continuous on $\mathbb{R}$.

- Piecewise linearity: The function consists of exactly two affine pieces with a single breakpoint at $u = 0$. On $(-\infty, 0)$ it is linear with slope $\alpha$, and on $[0, \infty)$ it is linear with slope 1. Therefore $\sigma$ is piecewise linear.

- $g_0 \in \mathcal{W}^{n,\infty}([0,1]^d)$ for all integers $n \geq 1$: recall that the Sobolev space $W^{n,\infty}(\mathcal{D})$ consists of functions whose partial derivatives up to total order $n$ are essentially bounded on $\mathcal{D}$. For each coordinate $j = 1, \ldots, p$, the first-order partial derivative of $g$ is

$$\frac{\partial g_0}{\partial x_j}(x) = \theta_{0j} \leq 1,$$

  which is constant and hence belongs to $L^\infty(D)$. Moreover, for any multi-index $k$ with $|k| \geq 2$, we have

$$D^k g_0(x) = 0 \quad \text{a.e. on } \mathcal{D},$$

  where $D$ is the derivative sign and therefore $D^k g_0 \in L^\infty(\mathcal{D})$. Since $\mathcal{D}$ is bounded, $g_0$ itself is also essentially bounded on $\mathcal{D}$, with

$$\|g_0\|_{L^\infty(\mathcal{D})} \leq \sum_{j=1}^p |\theta_{0j}|.$$

Consequently, all weak derivatives $D^k g_0$ with $|k| \leq n$ exist and belong to $L^\infty(\mathcal{D})$, implying that $g_0 \in W^{n,\infty}([0,1]^d)$ for all integers $n \geq 1$.

We next show that minimizing training loss is equivalent to minimizing the distance between $g(X; \beta)$ and $g_0(X)$. Define the population Fréchet risk

$$R(g) = \mathbb{E}\big[d^2(Y, \zeta(g(X; \beta)))\big].$$

Note that $R(g)$ is minimized at $g_0$, and for any $g$ with $R(g) \leq R(g_0) + \epsilon_n$ and $\epsilon_n \to 0$, we have

$$\mathbb{E}_X\big[(g(X; \beta) - g_0(X))^2\big] \to 0,$$

under Assumptions 1, 2 and 3. The minimum of $R(g)$ is achieved at $g_0$ by construction of the model (single-index structure). Suppose $R(g) - R(g_0) \leq \varepsilon_n \to 0$. If $\mathbb{E}_X[(g(X; \beta_\delta) - g_0(X))^2] \not\to 0$, then by Assumption 2 and compactness, the directions differ by a positive angle, implying $R(g) - R(g_0) \geq \eta > 0$ on a set of positive probability (from the density assumption), contradicting $\varepsilon_n \to 0$. Thus the index error must vanish.

The true index $g_0(x) = x^\top \theta_0$ is continuous on compact $\mathcal{D}$ with radius as $C_0$. By the universal approximation theorem by Yarotsky (2017), for any $\delta > 0$, there exists a DNN $g(\cdot; \beta_\delta)$ satisfying the bound. Then for every $\delta > 0$ there exists a Leaky ReLU neural network $g(\cdot; \beta_\delta)$ such that

$$\sup_{x \in \mathcal{D}}\big|g(x; \beta_\delta) - g_0(x)\big| < \delta,$$

where $\sup_{x \in \mathcal{D}}\big|g(x; \beta_\delta) - g_0(x)\big| = \sup_{x \in \mathcal{D}}\big|x^\top(\widehat{\theta} - \theta_0)\big| < \delta$. We can find the largest Euclidean ball with radius $C_0$ and at center $x_0 \in \mathbb{R}^p$ such that

$$x_0 + B_{C_0} := \{x_0 + u : \|u\|_2 \leq C_0\} \subset \mathcal{D},$$

where $B_{C_0}$ is a Euclidean ball around the origin with radius $C_0$. Since $x_0 + B_{C_0} \subset \mathcal{D}$, the two points

$$x_\pm := x_0 \pm C_0 \frac{\widehat{\theta} - \theta_0}{\|\widehat{\theta} - \theta_0\|_2}$$

belong to $\mathcal{D}$. Therefore,

$$\sup_{x \in \mathcal{D}} |x^\top(\widehat{\theta} - \theta_0)| \geq \max\{|x_+^\top(\widehat{\theta} - \theta_0)|, |x_-^\top(\widehat{\theta} - \theta_0)|\}.$$

A direct calculation yields

$$x_\pm^\top(\widehat{\theta} - \theta_0) = x_0^\top(\widehat{\theta} - \theta_0) \pm C_0\|\widehat{\theta} - \theta_0\|_2.$$

Let $a := x_0^\top \widehat{\theta} - \theta_0 \in \mathbb{R}$ and $b := C_0\|\widehat{\theta} - \theta_0\|_2 \geq 0$. Then

$$\max\{|x_+^\top(\widehat{\theta} - \theta_0)|, |x_-^\top(\widehat{\theta} - \theta_0)|\} = \max\{|a + b|, |a - b|\}.$$

We now use the elementary identity, valid for all $a \in \mathbb{R}$ and $b \geq 0$,

$$\max\{|a + b|, |a - b|\} = |a| + b.$$

It then gives

$$\sup_{x \in \mathcal{D}} |x^\top(\widehat{\theta} - \theta_0)| \geq |x_0^\top(\widehat{\theta} - \theta_0)| + C_0\|\widehat{\theta} - \theta_0\|_2 \geq C_0\|\widehat{\theta} - \theta_0\|_2.$$

Thus,

$$\|\widehat{\theta} - \theta_0\|_2 \leq \frac{1}{C_0} \sup_{x \in \mathcal{D}} |x^\top(\widehat{\theta} - \theta_0)| < \delta/C_0.$$

$\square$

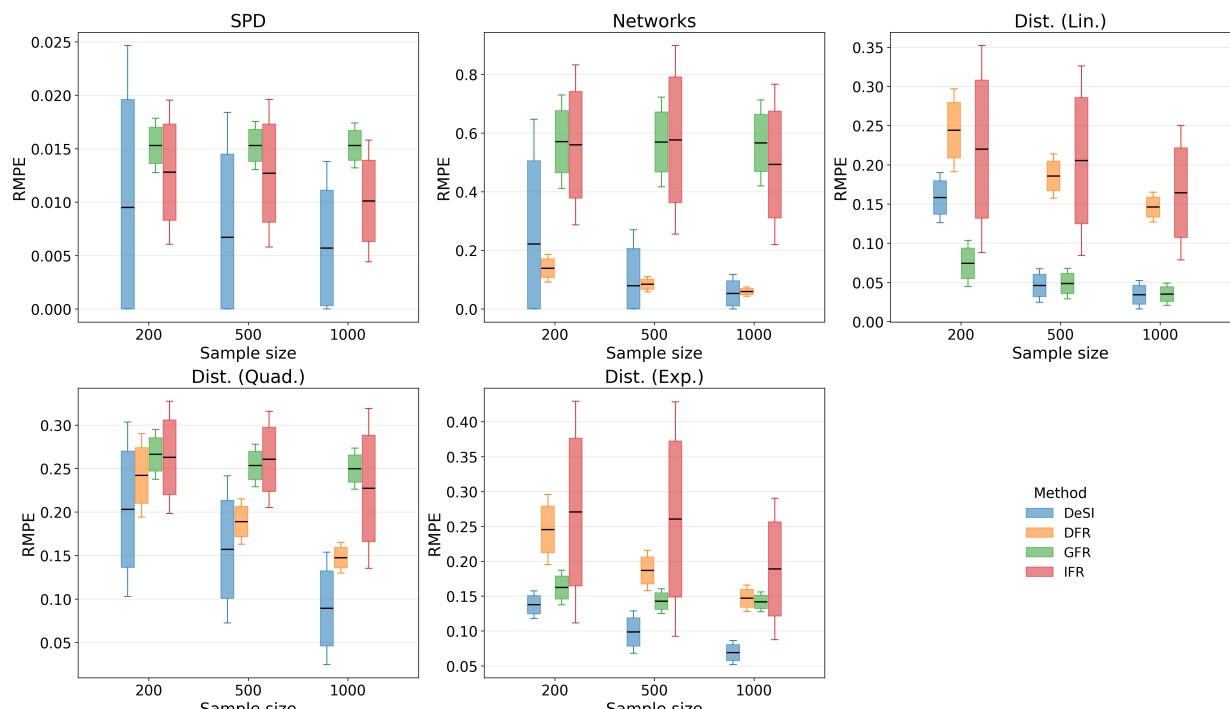

*Figure 4.* MPE across five different simulation settings. Boxplots summarize the mean and standard deviation of prediction errors over 200 Monte Carlo replications for each competing method. The panels display the results for SPD matrix outputs, network-valued outputs, and distribution-valued outputs under linear, quadratic, and exponential link functions. Across most settings, DeSI achieves lower MPE as the sample size increases, showing improved prediction accuracy with more training data. The legend in the bottom-right panel identifies the compared methods.

*Proof of Lemma 4.2.* Based on Theorem 4.1, we can assume that

$$\|\widehat{\theta} - \theta_0\|_2 = O_p(b_n),$$

and the proof is a direct result from Iao et al. (2025), where the dimension is one. □

*Proof of Theorem 4.3.* Let $z = g_0(x)$ be the true (unknown) index value for the new test point $x$, and let $\widehat{z} = g(x; \beta_\delta)$ be the estimated index produced by the fitted deep network.

We decompose the prediction error as follows:

$$d\big(\widehat{\zeta}_h(\widehat{z}; \{\widehat{Z}_i, Y_i\}_{i=1}^n), \zeta(z)\big)$$
$$\leq d\big(\widehat{\zeta}_h(\widehat{z}), \widetilde{\zeta}_h(\widehat{z})\big) + d\big(\widetilde{\zeta}_h(\widehat{z}), \widetilde{\zeta}_h(z)\big) + d\big(\widetilde{\zeta}_h(z), \zeta(z)\big), \tag{8}$$

where $d\big(\widehat{\zeta}_h(\widehat{z}), \widetilde{\zeta}_h(\widehat{z})\big)$ and $d\big(\widetilde{\zeta}_h(\widehat{z}), \widetilde{\zeta}_h(z)\big)$ are bounded by $O_p\big\{\big(h^{-2}b_n\big)^{1/(\gamma_3-1)}\big\}$ under Lemma 4.2, and the last term is bounded by $O_p\big\{h^{2/(\gamma_1-1)} + (nh)^{-1/(2\gamma_2-2)}\big\}$ in Petersen & Müller (2019), which is the convergence rate for empirical local Fréchet regression estimator to the true conditional Fréchet mean. □

# E. Additional Plots for Simulations

Figure 4 summarizes the predictive performance of DeSI and competing methods across all simulation settings. For each response space, we report the MPE at sample sizes 200, 500, and 1000. The results show that DeSI generally improves as the sample size increases and is competitive or favorable across heterogeneous output types, supporting the flexibility of the proposed DeSI framework.

Figure 5 visualizes the estimation accuracy for the single index $\theta_0$. Across all simulation settings, DeSI yields smaller MPE than IFR, indicating more accurate recovery of the underlying single-index structure. The advantage is especially

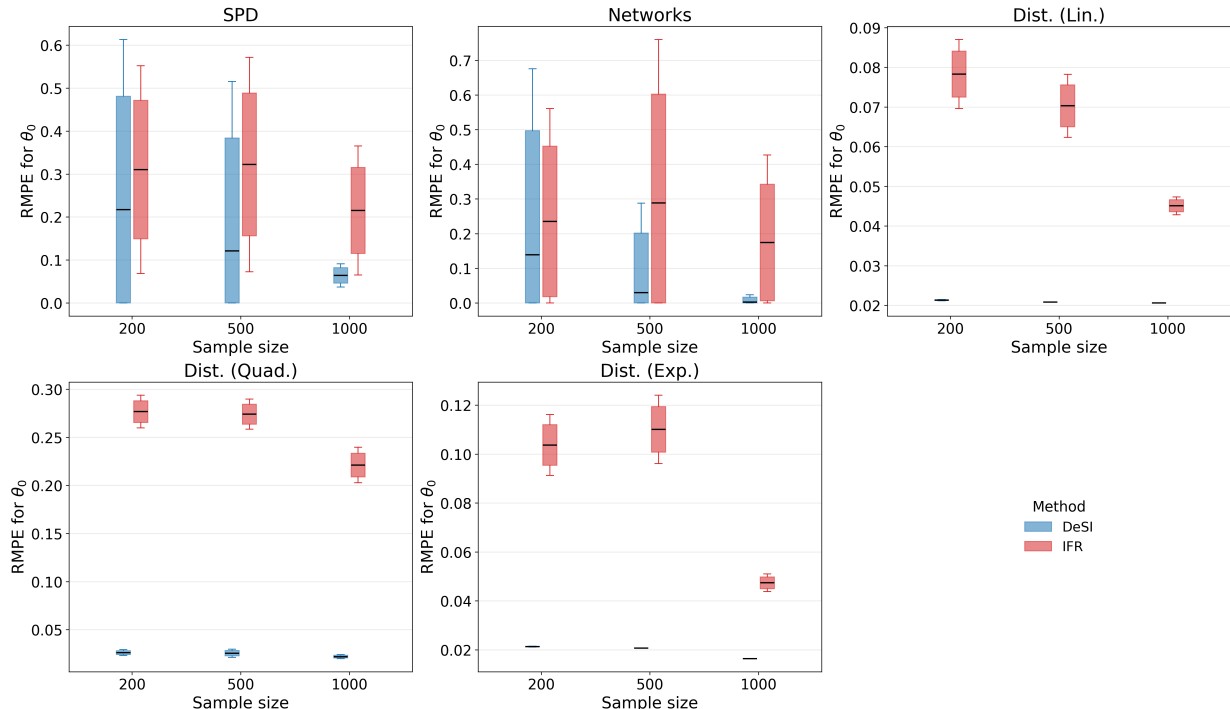

*Figure 5.* MPE for estimating the true single-index direction $\theta_0$ across five simulation settings. The boxplots display the mean and standard deviation over 200 Monte Carlo replications for DeSI and IFR. The panels provide results for SPD matrix outputs, network-valued outputs, and distribution-valued outputs under linear, quadratic and exponential links. Lower values indicate more accurate recovery of the single index.

pronounced in the network and distributional settings, where the DeSI estimates remain stable as the sample size increases. The proposed DeSI framework improves not only output prediction but also estimation of the latent index direction.

## F. Robustness Beyond the Linear Single-index Setting

To investigate robustness, we considered a data-generating mechanism that departs from the single-index assumption. Specifically, we generated distributional outputs as normal distributions

$$Y_i = N(\mu_i, 1),$$

where

$$\mu_i \sim N\left(X_{i1}^2 + X_{i2}^2 + X_{i3}^2 + X_{i4}^2, 0.25\right),$$

and the inputs $X_i$ were generated as in Section 5.3. This corresponds to an additive model with component functions $f_j(x) = x^2$ for $j = 1, \ldots, 4$. We compared four settings: GFR with input $X_i$, GFR with input $X_i^2$, DeSI with input $X_i$, and DeSI with input $X_i^2$. The MPEs based on 100 repeated runs are reported in Table 4.

*Table 4.* MPEs for the additive model robustness experiment. Standard deviations over 100 repeated runs are shown in parentheses.

| Method | $n = 200$ | $n = 500$ | $n = 1000$ |
|---|---|---|---|
| GFR with input $X$ | 0.5889 (0.0270) | 0.5905 (0.0181) | 0.5872 (0.0252) |
| GFR with input $X^2$ | 0.0599 (0.0225) | 0.0400 (0.0146) | 0.0284 (0.0092) |
| DeSI with input $X$ | 0.3977 (0.0414) | 0.3765 (0.0367) | 0.3810 (0.0329) |
| DeSI with input $X^2$ | 0.0731 (0.0230) | 0.0491 (0.0211) | 0.0439 (0.0149) |

When using $X_i^2$ as the input, GFR is correctly specified and consequently achieves the best overall performance, as expected. Under this ideal setting, DeSI does not outperform GFR. In contrast, when both methods are applied to the original input $X_i$, DeSI consistently outperforms GFR, suggesting greater robustness when the underlying model structure is misspecified.

## G. Computational Cost and Sensitivity Analysis

The average training times, measured in seconds, in the network-valued setting are reported in Table 5.

*Table 5.* Average training time in seconds for the network-valued setting.

| Method | $n = 200$ | $n = 500$ | $n = 1000$ |
|--------|-----------|-----------|------------|
| DeSI | 1.20 s | 1.40 s | 5.93 s |
| IFR | $\sim 2$ h | $\sim 3$ h | $\sim 5$ h |
| GFR | 0.83 s | 1.16 s | 2.76 s |
| DFR | 23.72 s | 71.31 s | 181.88 s |

While GFR is the most computationally efficient method and is competitive with DeSI in terms of training time, it typically has lower predictive accuracy because it is much more restrictive, similar to linear regression. A major additional disadvantage of GFR compared with DeSI is that it does not provide the same level of interpretability. Compared with IFR and DFR, DeSI is substantially more efficient in practice, while still achieving strong predictive performance and retaining the interpretability of the single-index structure.

We also investigated the stability of jointly optimizing the bandwidth and DNN parameters through a sensitivity analysis in the case of network-valued outputs. Specifically, we conducted 100 repeated runs with bandwidth initializations ranging from 0.1 to 0.9. The mean prediction errors and corresponding standard deviations are summarized in Table 6.

*Table 6.* Sensitivity analysis for bandwidth initialization in the network-valued setting. MPEs are reported with standard deviations over 100 repeated runs shown in parentheses.

| Initial bandwidth | $n = 200$ | $n = 500$ | $n = 1000$ |
|-------------------|-----------|-----------|------------|
| 0.1 | 0.0716 (0.0517) | 0.1071 (0.1640) | 0.0373 (0.0142) |
| 0.3 | 0.1188 (0.0993) | 0.0515 (0.0334) | 0.0539 (0.0394) |
| 0.5 | 0.0999 (0.0525) | 0.0497 (0.0258) | 0.0562 (0.0283) |
| 0.7 | 0.2813 (0.3074) | 0.0391 (0.0144) | 0.0353 (0.0116) |
| 0.9 | 0.1572 (0.1875) | 0.0438 (0.0176) | 0.0470 (0.0238) |

Overall, the model is reasonably stable across a wide range of bandwidth initializations, especially for moderate and large sample sizes. More notable variability is observed for small $n$, which is expected due to limited data. In practice, we also include a mild regularization on the bandwidth to prevent degenerate solutions.

## H. Choice of Hyperparameters

The hyperparameters for DeSI can be selected using a grid search over the candidate values listed in Table 7. The optimal combination of hyperparameters is chosen to minimize the mean prediction error for the validation data.

*Table 7.* Hyperparameter settings.

| | | | | | |
|---|---|---|---|---|---|
| $\lambda$ | 0.0005 | 0.0010 | 0.0050 | 0.0100 | 0.0500 |
| Dropout rate | 0.1500 | 0.2000 | 0.2500 | 0.3000 | 0.3500 |
| Learning rate | 0.0010 | 0.0050 | 0.0100 | 0.0500 | 0.1000 |
| Leaky ReLU slope $\alpha$ | 0.0050 | 0.0100 | 0.05000 | 0.1000 | 0.2000 |
| Number of hidden layers | 2 | 3 | 4 | 5 | 6 |
| Number of neurons | 8 | 16 | 32 | 64 | 128 |

