# OpenReview forum: "Deep Single-Index Fréchet Regression"
_ICML.cc/2026/Conference — ICML 2026 regular_

### Official Review · Reviewer_a8ay · 2026-03-12

**Soundness:** 3
**Presentation:** 3
**Significance:** 4
**Originality:** 3
**Overall Recommendation:** 4
**Confidence:** 3

**Summary:**

This paper proposes DeSI (Deep Single-Index Frechet Regression) to tackle the regression problem for high-dimensional inputs and non-Euclidean metric space outputs. To solve the curse of dimensionality and the lack of interpretability in existing deep learning methods, DeSI uses a deep neural network to project multivariate inputs into a 1-dimensional single-index, and then directly performs local Frechet regression along this estimated index

**Compliance With Llm Reviewing Policy:**

Affirmed.

**Final Justification:**

Please see my "Rebuttal Acknowledgement"

**Key Questions For Authors:**

In Algorithm 1, local Frechet regression  is calculated for every training epoch to get the predicted output. Since LFR does not always have a closed-form solution, I am wondering if the training time is too slow for large datasets. Providing the theoretical time complexity or actual training time compared to baseline models will improve the soundness evaluation.

The bandwidth $h$ is treated as a learnable parameter and optimized jointly with the DNN parameters $\beta$. Does this joint optimization suffer from instability or local minima, and how sensitive is the model to the initialization of $h^{(0)}$? Clarifying the stability of this optimization will resolve concerns about the methodological design.

The DeSI framework heavily depends on the assumption of a single-index structure. If the data is highly complex, one index dimension might not be enough to capture the variance. Can this framework be extended to a multi-index model? If the authors briefly discuss the potential for a multi-index extension, it will positively change the significance score.

**Limitations:**

yes

**Strengths And Weaknesses:**

<Strengths>
Soundness: The technical soundness is very strong. The authors provide solid mathematical proofs for the consistency of the estimated index and the convergence rate of the estimator. Furthermore, the empirical experiments are well-designed and clearly support the claim that DeSI outperforms existing baseline models.

Presentation: The paper is well-structured and logical. The narrative flows nicely, and the architecture diagrams (Figure 1 and Figure 2) are very helpful for visually understanding the two-module framework.

Significance: The paper addresses a highly relevant and growing problem: performing regression with high-dimensional inputs and non-Euclidean outputs. This provides great practical utility for fields dealing with complex modern data structures.

Originality: The approach of combining deep neural networks with local Frechet regression is highly creative. It brilliantly tackles the curse of dimensionality while keeping the model interpretable, which is a known weakness of standard deep learning.

<Weaknesses>
Soundness & Originality: While the theoretical guarantees are strong, they rely heavily on standard but very strict mathematical assumptions (Assumptions A1-A5) regarding local curvature and smoothness. It is slightly questionable how well these assumptions hold up in extremely noisy or irregular real-world data outside of the provided New Jersey mood dataset.

Presentation: The mathematical density in the theoretical sections and appendix is very heavy. Adding slightly more intuitive, high-level explanations for the complex equations could make the paper much more accessible to general machine learning practitioners.

---

> ### Author Rebuttal · Authors · 2026-03-30
>
> We sincerely thank the reviewer for the thoughtful and constructive comments on the computational efficiency, optimization stability, and possible extensions of our framework. We are encouraged that these questions focus on the practical and methodological aspects of DeSI, and we appreciate the opportunity to provide further clarifications.  Below we address each point in turn.
>
> - **Training time and computational cost of Algorithm 1.**
>
>   Thank you for raising this important question. Although local Fréchet regression is evaluated at each training epoch, the empirical training time of DeSI remains competitive in practice. The average training times (in seconds) in the network-valued setting are:
>
>     |Method|$n=200$|$n=500$|$n=1000$|
>     |-|-:|-:|-:|
>     |DeSI|1.20 s|1.40 s|5.93 s|
>     |IFR|~2 h|~3 h|~5 h|
>     |GFR|0.83 s|1.16 s|2.76 s|
>     |DFR|23.72 s|71.31 s|181.88 s|
>
>   While GFR is the most computationally efficient method and is competitive compared to DeSI, it typically has lower predictive accuracy as it is much more restrictive, similar to linear regression. A major additional disadvantage compared to DeSI is that it does not provide interpretability. Compared with IFR and DFR, DeSI is substantially more efficient in practice, while still achieving strong predictive performance and retaining the interpretability of the single-index structure. We will include these results along with a brief additional discussion of computational complexity  in the revised manuscript.
>
> - **Stability of joint optimization with learnable bandwidth.**
>
>   Thank you for this insightful question. We investigated the stability of jointly optimizing the bandwidth and DNN parameters through a sensitivity analysis in the case of network outputs. Specifically, we conducted 100 repeated runs with bandwidth initializations ranging from $0.1$ to $1.0$. The mean squared prediction errors and corresponding standard deviations are summarized below:
>
>     |Initial bandwidth|$n = 200$|$n = 500$|$n = 1000$|
>     |-|-:|-:|-:|
>     |0.1|0.0716 (0.0517)|0.1071 (0.1640)|0.0373 (0.0142)|
>     |0.3|0.1188 (0.0993)|0.0515 (0.0334)|0.0539 (0.0394)|
>     |0.5|0.0999 (0.0525)|0.0497 (0.0258)|0.0562 (0.0283)|
>     |0.7|0.2813 (0.3074)|0.0391 (0.0144)|0.0353 (0.0116)|
>     |0.9|0.1572 (0.1875)|0.0438 (0.0176)|0.0470 (0.0238)|
>     |1.0|0.1745 (0.2413)|0.0723 (0.0341)|0.0826 (0.1261)|
>
>   Overall, the model is reasonably stable across a wide range of initializations, especially for moderate and large sample sizes. More notable variability is observed for small $n$, which is expected due to limited data. In practice, we also include a mild regularization on the bandwidth to prevent degenerate solutions. We will include this sensitivity analysis in the revised manuscript to clarify the stability of the optimization procedure.
>
> - **Extension beyond the single-index model.**
>
>   We appreciate this forward-looking suggestion. We agree that when the relationship between inputs and outputs is highly complex, a single index may not be sufficient to capture all relevant variation. In principle, our framework could indeed be extended to a multi-index model, in which local Fréchet regression is performed on multiple learned indices rather than only one. Such an extension could increase modeling flexibility and potentially improve predictive performance. However, this comes with a trade-off: one of the main strengths of DeSI is its interpretability through a single global direction, whereas in the multi-index setting, multiple directions must be interpreted jointly, making the contribution of each predictor less transparent. We agree that discussing this extension would strengthen the significance of the work, and we will add a brief discussion of the multi-index extension and the interpretability-flexibility trade-off in the revised manuscript.
>
> We sincerely thank you again for your constructive suggestions, which will help us further improve both the clarity and the practical relevance of the paper.

---

> > ### Author Rebuttal · Reviewer_a8ay · 2026-04-03
> >
> > Thank you for the thorough responses. The training time comparison and bandwidth sensitivity analysis are convincing and address my concerns well. I maintain my original score, but will increase my confidence score.

---

### Official Review · Reviewer_gxMx · 2026-03-13

**Soundness:** 4
**Presentation:** 4
**Significance:** 3
**Originality:** 3
**Overall Recommendation:** 5
**Confidence:** 3

**Summary:**

This paper studies regression where the outputs can be in general metric spaces, rather than Euclidean spaces.

Background/Definitions
- Frechet regression is the generalization of Euclidean regression to general metric spaces.
- Given a random variable Y in a metric space, which depends on another variable X, the goal of Frechet regression is to estimate the Frechet mean of Y given X.
    - Here, the Frechet mean is a generalization of expected value to metric spaces.
- Local Frechet regression (LFR), introduced by previous work, is a generalization of locally-weighted linear regression, using a kernel to estimate the distance between inputs.
- LFR takes in a scalar-valued input.

Deep Single-Index Frechet Regression (DeSI)
- This method assumes that the target function has a single-index structure, meaning that for an independent variable X, the output is purely a function of $\theta_0^T X$ for some vector $\theta_0$.
- This vector $\theta_0$ is in fact a function of X, and is estimated in this work using a multi-layer MLP.
- The MLP projects each input vector X into $\mathbb{R}$, and then LFR is applied to these projections to estimate the targets $Y_i$ (given a dataset of data points X and labels Y, the single-index MLP is applied to the data points X prior to applying LFR).
- The bandwidth of the kernel, relevant during the LFR stage, is treated as a learnable parameter, and optimized together with the parameters of the single-index MLP.
    - The loss function penalizes extremely small bandwidths, to prevent the LFR stage from overfitting.
- During the training procedure, LFR is applied to each point in the dataset, in a leave-one-out manner.
- The paper gives theoretical results on the convergence rates of DeSI.

Simulations
- They conduct simulations where the targets are in various metric spaces (PSD matrices, graphs using the Frobenius norm distances between Laplacians, and 1D probability distributions with Wasserstein distance).
    - In all cases, the ground-truth is generated by a single-index model, where the single-index direction is fixed across examples.
    - They report the prediction error, defined as either (A) the error in the regression output on test examples, or (B) the error in predicting the ground-truth single-index vector.
    - In all settings, DeSI has lower prediction error than baselines, across different sample sizes.

**Compliance With Llm Reviewing Policy:**

Affirmed.

**Final Justification:**

I am increasing my score to 5, as my initial confusion about the interpretability benefits of single-index models was resolved by the authors in the rebuttal. The writing is otherwise clear and the results seem strong.

**Key Questions For Authors:**

- Could you clarify whether, in the single-index model, $\theta_0$ is a function of X, or is independent of X? Since $\theta_0$ is estimated by a deep neural network given X, it seems that it is a function of X? In the New Jersey mood dataset, why is $\theta_0$ a fixed vector that is constant across the data?
    - According to lines 212-219 on the left column of page 4, it seems that $\theta_0$ is a function of X rather than being a fixed vector.
- In Assumption 2 of Section 4, could you explain the role of $\theta \neq c \theta_0$? Why not have $\theta \neq \theta_0$?
- In the simulations, in the graph Laplacian setting, could you clarify what is $X_i$ in the definition of the weighted adjacency matrix E? How does E depend on the single-index direction $\theta_0$?
- Tables 1 and 2 may be easier to read if they are shown as graphs, where the x-axis represents the sample size.
- In Table 2, could you clarify why the standard deviation is larger than the mean, for DeSI in the SPD setting? Additionally, why does the performance worsen from 500 to 1000 samples?
- Could you compare and contrast your method with DFR, and clarify the advantages?
- How does DFR perform on the mood data?

**Limitations:**

yes

**Strengths And Weaknesses:**

Strengths
- Paper is clearly written and methodology seems sound.
- The experimental results in Table 1 and Table 2 seem strong, as DeSI outperforms other methods for Frechet regression in prediction error.

Weakness
- The interpretability benefit of single-index model sounds unclear. Since $\theta_0$ can be a function of x, in theory, the expressivity of the single-index model sounds equal to the expressivity of an arbitrary model that maps the input x to a scalar.

---

> ### Author Rebuttal · Authors · 2026-03-30
>
> We sincerely thank you for your thoughtful and careful review. We are glad that you found the methodology **sound** and the experimental results **strong**. Your questions are very helpful for improving the clarity of the paper, and we address each point below.
>
> - **Clarification of the single-index model.**
>
>   Thank you for raising this important question. We apologize for the confusion in the current presentation. In our model, the true single-index direction $\theta_0$ is independent of $X$ and is a fixed population-level vector, as in the standard single-index framework. The model assumes $m(X) = \zeta(X^\top \theta_0)$. The role of the DNN is not to make $\theta$ a function of $X$, but to estimate the global direction $\theta_0$ in a flexible, data-adaptive way. In implementation, the DNN outputs a vector in $\mathbb{R}^p$, which is normalized to unit length and interpreted as $\theta$. The subject-specific estimates $\hat{\theta}_i$ arise from evaluating the DNN at different inputs during training. However, these should be viewed as sample-level estimates that concentrate around a single underlying direction $\theta_0$, rather than fundamentally different directions. In practice, we aggregate these estimates via their intrinsic (Fréchet) mean on the unit sphere to obtain a global estimate. We agree that this was not clearly explained in the paper (especially around lines 212–219), and we will revise the manuscript to make this distinction explicit.
>
> - **Assumption 2 clarification.**
>
>   Thank you for the careful reading. You are correct that the notation involving $c$ is unnecessary in its current form. The purpose of Assumption 2 is to ensure identifiability of the index direction (up to scale/sign). We will simplify and correct this condition in the revised manuscript.
>
> - **Definition of the weighted adjacency matrix in the graph Laplacian simulation.**
>
>   We are grateful for this important comment. In the simulation, the quantities entering the construction should depend on the scalar index $Z_i = X_i^\top \theta_0$, rather than the full vector $X_i$. The current description contains a typo; we will correct it in the revision.
>
> - **Presentation of Tables 1 and 2.**
>
>   Thank you for this helpful suggestion. We agree that visualizing performance as a function of sample size would make trends easier to interpret. At the same time, the tables provide precise numerical comparisons across multiple methods, metric spaces, and settings. In the revision, we will retain the tables for completeness and additionally include plots with sample size on the x-axis, which will make performance trends more transparent while preserving detailed comparisons.
>
> - **Table 2: large variance and behavior at $n=1000$.**
>
>   We sincerely thank you for the careful examination of Table 2 and for raising this important point. Upon revisiting the table, we found that this issue was due to a typo. The correct result for $n=1000$ is 0.0640 (0.0181). After correcting this, the results show the expected improvement as the sample size increases, and the standard deviation becomes smaller relative to the mean for larger $n$. For smaller sample sizes, the relatively large variability likely reflects the instability in estimating the index direction when data are limited.
>
> - **Comparison with DFR and its advantages.**
>
>   Thank you for this important question. Conceptually, DFR and DeSI differ in their model structure. DFR employs neural networks to map Euclidean inputs to a low-dimensional manifold representation of the metric space-valued output, thereby relying on an assumption of a low-dimensional underlying manifold. In contrast, DeSI imposes a single-index structure, where the model depends on $X$ through a scalar projection $Z = X^\top \theta$, with a global direction $\theta$ independent of $X$. This combines the flexibility of DNNs for learning the index with the interpretability of the single-index model: the components of $\theta$ directly quantify the contribution of each predictor, which is a distinguishing feature of DeSI  that is not available in DFR. We will revise the manuscript to clarify this distinction and will highlight when each approach may be preferable.
>
> - **DFR on the mood data.**
>
>   Thank you for this suggestion. We did not include DFR on the mood dataset because the current implementation of DFR does not support compositional outputs, and extending it would require nontrivial adaptations that depend on the specific metric space and distance. We will clarify this limitation more explicitly in the revision.
>
> We sincerely thank you again for your careful reading and constructive suggestions. Your feedback will help us significantly improve both the clarity and presentation of the paper.

---

> > ### Author Rebuttal · Reviewer_gxMx · 2026-04-04
> >
> > I appreciate the detailed response from the authors. I have a follow-up question again regarding the expressivity of the single-index model.
> >
> > How do you ensure that the subject-specific estimates concentrate around the underlying direction? Is this just because the underlying tasks are represented by a single-index model, and therefore Table 2 demonstrates that all the subject-specific estimates are close to the ground-truth?
> >
> > Additionally, if the target task no longer has a single-index structure, then DeSI will still be able to attain low error, but the claim that "subject-specific estimates concentrate around the underlying direction" will no longer hold - is this understanding correct?

---

> > > ### Author Response · Authors · 2026-04-04
> > >
> > > We sincerely thank you for this thoughtful follow-up question, which helps clarify an important aspect of the model.
> > >
> > > The concentration of the subject-specific estimates $\hat{\theta}_i$ around the true direction $\theta_0$ is fundamentally tied to the **single-index structure** of the data-generating mechanism. Under this assumption, the regression function depends on $X$ only through the scalar projection $Z = X^\top \theta_0$. As a result, any direction that deviates substantially from $\theta_0$ will lead to a systematic loss in predictive accuracy.
> > >
> > > Since our training procedure minimizes the Fréchet regression loss, this induces a form of **implicit alignment**: directions that are closer to $\theta_0$ yield better predictions, and therefore are favored by the optimization. Consequently, the learned directions $\hat{\theta}_i$ concentrate around $\theta_0$. This is also reflected empirically in Table 2, where the estimation error of the index direction decreases as the sample size increases.
> > >
> > > You are also correct that this concentration property relies on the single-index assumption. When the true data-generating mechanism follows a single-index model, $\theta_0$ is identifiable (up to sign), and consistent estimation is expected.
> > >
> > > When this assumption is violated, the situation changes. In that case, there may not exist a single direction that fully captures the dependence structure, and thus the notion of concentration around a true $\theta_0$ no longer strictly applies. However, DeSI still seeks the **best one-dimensional projection in terms of predictive performance**, and can achieve low prediction error by approximating the underlying structure. This is consistent with our additional robustness experiments, where DeSI performs well even under model misspecification.
> > >
> > > We will clarify this distinction in the revised manuscript to make explicit that:
> > > - concentration of $\hat{\theta}_i$ around $\theta_0$ holds under the single-index assumption, and
> > > - under misspecification, the method instead recovers an optimal projection rather than a true underlying direction.

---

### Official Review · Reviewer_gtTb · 2026-03-14

**Soundness:** 2
**Presentation:** 3
**Significance:** 4
**Originality:** 3
**Overall Recommendation:** 5
**Confidence:** 3

**Summary:**

The paper proposes a method for regression with metric-space-value: the core idea is to reduce a high-dimensional euclidean input to a learned one-dimensional index, then apply local frechet regression along that. The authors claim that this is a good compromise between flexibility and interpretability (since we have the single index). The paper also provides theory plus experiments on simulated output geometries and a real compositional dataset. Since classical local frechet regression suffers from the curse of dimensionality and existing deep approaches for object-valued outputs are often less interpretable, this is an interesting idea! The empirical results are promising.

**Compliance With Llm Reviewing Policy:**

Affirmed.

**Key Questions For Authors:**

Theorem 4.1: I want to make sure I understood this right. It says "Consistency of the estimated direction" but the statement is existential, saying that for any delta there exists a sufficiently expressive Leaky-ReLU network approximating g uniformly, from which closeness of theta-cap is inferred? Is that a consistency result for the trained estimator? (Pls tell me if I am being stupid or missing something here.) I think the proof snippet is super short (maybe you cut due to page limits) but I am having a hard time understanding this.

It looks like the simulations bit says that one gets subject specific ests theta-cap_i which are avg on the sphere? Is that correct? Do we get a sample-dependent direction or a global index dir?

**Limitations:**

Yes

**Strengths And Weaknesses:**

Critically, this is a super important problem (regression with non-euclidean stuff is a pain) and the overall "design" is really elegant: learning a 1D index! The paper does appropriate experiments and checks interpretability via the estimated direction.

The loss depends on leave-one-out local frechet regression predictions at every epoch, and the parameters are updated with grad-opt. But the paper does not clearly explain how gradients are computed through the argmin in the frechet regression layer across the different metric spaces, or what assumptions are needed for differentiability/stability.

My main concern is that there seem to be a lot of missing pieces which made this quite hard to read in places. I suspect this is due to the authors cutting out things to fit within page limits. But it led to a lot of confusion.

---

> ### Author Rebuttal · Authors · 2026-03-30
>
> We sincerely thank you for the positive assessment of our work and for the careful and thoughtful feedback. We are especially encouraged that you found the design **elegant** and the empirical results **promising**. We address all of your questions in detail below.
>
> - **Gradient computation through the Fréchet regression layer.**
>
>   Thank you for raising this important point. Due to space limitations, the details of gradient computation were moved to the appendix. Specifically, Appendix B provides the gradient calculations for the metric space-valued outputs considered in the paper, including SPD matrices under the log-Cholesky metric, networks under the Frobenius metric, compositional data under the geodesic distance, and univariate probability distributions under the Wasserstein metric. The required assumptions are standard, namely existence and uniqueness of the Fréchet mean together with local smoothness of the objective. We agree that this is an important implementation detail, and in the revision we will include a concise summary in the main text and provide a clearer reference to Appendix B.
>
> - **Missing details and clarity of presentation.**
>
>   Thank you for this important feedback. We agree that some details were overly compressed due to page limits, which may have made parts of the paper harder to follow. In the revision, we will improve readability by:
>   - adding a clearer description of the DNN module and its output,
>   - explicitly defining $\theta$ and its role,
>   - summarizing gradient computation in the main text,
>   - and providing more explicit pointers to the appendix.
>
> - **Theorem 4.1.**
>
>   You are absolutely right in your reading, and your interpretation is correct. The current title is indeed misleading. The theorem is not a standalone statistical consistency result, but rather an approximation result showing that sufficiently expressive DNNs can uniformly approximate the true index function. We will rename the theorem to “Uniform approximation of the estimated direction” and revise the presentation to clearly distinguish between approximation and statistical consistency. We will also expand the proof to provide more detail and improve clarity.
>
> - **Subject-specific $\theta_i$.**
>
>   Thank you for this insightful question. The underlying model assumes a global index direction $\theta_0$, shared across all subjects, as in the standard single-index framework. The subject-specific estimates $\hat{\theta}_i$ arise because the DNN is evaluated at different inputs $X_i$, which leads to small variations in the estimated direction across samples during training. Importantly, these $\hat{\theta}_i$ should be viewed as noisy estimates of the same underlying direction $\theta_0$, rather than fundamentally different directions. In our simulations, we indeed observe that the $\hat{\theta}_i$ are distributed around the true $\theta_0$, indicating that each individual estimate provides a reasonable approximation of the underlying index direction. To obtain a single estimator, we aggregate $\\{\hat{\theta}\_i\\}\_{i=1}^n$ using their intrinsic (Fréchet) mean on the unit sphere (as the length of the direction vectors is normalized to 1),  which yields a stable global estimate of $\theta_0$. We agree that the current presentation may give the impression of subject-specific parameters, and we will revise the manuscript to clarify that the model targets a global direction, while the $\hat{\theta}_i$ serve as sample-level estimates that concentrate around $\theta_0$.
>
> We sincerely thank you again for your careful reading and constructive suggestions. Your feedback will help us significantly improve both the clarity and presentation of the paper.

---

> > ### Author Rebuttal · Reviewer_gtTb · 2026-04-04
> >
> > I have understood the comments and stand by my score.

---

### Official Review · Reviewer_Q3xz · 2026-03-18

**Soundness:** 2
**Presentation:** 2
**Significance:** 4
**Originality:** 2
**Overall Recommendation:** 5
**Confidence:** 4

**Summary:**

The authors  proposes a method they coin DeSI for a regression task with the response taking value in a general metric space.
Predictors live in R^p and are potentially high-dimensional.
To circumvent the curse of dimensionality, DeSI is composed of two modules:
The first module is a feed-forward neural network that maps the predictors to a single real-valued index.
The second module applies local Frechet Regression (the generalisation of local linear regression to respones in metric spaces) to the single index.
Optimisation of the two modules is done jointly, i.e. both the weights of the neural network and the bandwidth are learnt via gradient descent.
The paper claims to provide a consistency result and convergence rates for DeSI.
Lastly, the is an empirical study consisting of 5 simulation scenarios and one real data set where DeSI show superior performance to the considered competitors.

**Compliance With Llm Reviewing Policy:**

Affirmed.

**Final Justification:**

The authors have clarified things that where previously not clear and I expect them to add those clarifications to the paper.
They also added a comparison to additive models which the submitted paper was lagging.

**Key Questions For Authors:**

- Can you make the theoretical contribution more clear and honest and have a broader literature review?
- Can you clarify your point in interpretability?
- Can you clarify what theta is/ what the DNN is doing?
- The empirical study is too narrow. Is it possible to compare to an additive model? What about a simulation with no single index structure, (sparse) high dimensional setting?  Is there more than one data set one can consider?

I am happy to increase my evaluation if these points are sufficiently addressed.

**Limitations:**

yes

**Strengths And Weaknesses:**

$\bf{Soundness}$ : The method is sound. However the theoretical analysis is much weaker than claimed. I would expect that much of the error is driven by the first module of DeSI. However for the first module the authors just re-state the well-known universal approximation property of feed-forward neural networks. The claim that Theorem 4.1 (the universal approximation property) implies consistency of the first module is plainly wrong. Convergence rates are shown under the assumption that the error of the first module is of certain order. However, under this assumption Theorem 4.2 and 4.3 are rather extensions of standardlocal Frechet regression arguments conditional on first-stage index-estimation error.
Lastly, I don't understand why DeSI is interpretable. Maybe this has something to do with theta (see below). This seems like a strong statement not further elaborated on in the paper. Does that imply  that your flexible feed-forward neural network is interpretable?

$\bf{Presentation}$: Presentation is mostly clear. However, there is a problem where it is not clear what the first module (the DNN) is doing. In the problem formulation g_0 is assumed linear (why?). But I guess the DNN learns a general function g_0 R^P->R. In Figure 2 and also later from the empirical study it seems that some theta is constraint to have norm 1. But what is theta (is it the last hidden layer? What dimension does it have?) It does not pop up in Section 3.2 when the DNN is explained. Later in Section 5 it seems that theta is individual specific...
Lastly, important literature around additive/partly linear additive models is missing. I am not sure how practical it is to implement them (this should be discussed in the paper if an empirical comparison is not feasible), but they must be mentioned as they solve exactly the same problem as the authors try to address, namely the curse of dimensionality -- even if they are mostly not as general. Here are examples of papers missing:
- Lin, Z., Müller, H. G., & Park, B. U. (2023). Additive models for symmetric positive-definite matrices and Lie groups. Biometrika, 110(2), 361-379.
- Han, K., Müller, H. G., & Park, B. U. (2020). Additive functional regression for densities as responses. Journal of the American Statistical Association.
- Jeon, J. M., & Park, B. U. (2020). Additive regression with Hilbertian responses. Ann. Statist. 48(5): 2671-2697 (October 2020). DOI: 10.1214/19-AOS1902
- Jeon, J. M., & Van Bever, G. (2025). Additive regression for Riemannian functional responses. Journal of Multivariate Analysis, 210, 105466.

$\bf{Significance}$: The paper tackles a interesting and "hot" topic. While the contribution is not super original it is solving a real weakness of current methods in regression for responses in general metric values spaces.

$\bf{Original}$: DeSI a logical extension/implementation of known concepts.  However it is interesting to see how much of a  difference this method makes in some empirical settings; e.g. those considered in the paper. However, more general cases could be considered.

---

> ### Author Rebuttal · Authors · 2026-03-30
>
> We sincerely thank you for your thoughtful and constructive feedback, and we are glad that you find the problem **important** and the method **sound**. We address your concerns below.
>
> - **Theoretical contribution and clarity.**
>
>   We agree that the current title of our first theorem is not fully accurate and will change it to  "Uniform approximation of the estimated direction" to better reflect its actual contents. More broadly, our theoretical contribution has two components. First, we combine a universal approximation argument for the DNN module with the single-index Fréchet regression framework to justify the approximation of the index function. Second, in Theorems 4.2–4.3, our contribution is to quantify how first-stage index estimation error propagates into the final estimator, which is specific to our end-to-end framework and does not arise in standard local Fréchet regression.
>
> - **Interpretability**
>
>   The interpretability in our framework does not come from the DNN itself, but from the *learned single-index direction $\theta$*. In our implementation, the DNN outputs a vector that is normalized to unit length. The regression model then depends on $X$ through the scalar projection $Z=X^\top\theta$, so the magnitude of each component of $\theta$ reflects the *relative importance* of the predictors. This is a key advantage of the single-index structure, and it is particularly valuable in our setting, since regression with metric space-valued outputs typically lacks interpretability due to the absence of linear operations. Our approach combines the flexibility of DNNs with the interpretability of single-index models.
>
> - **Role of the DNN, $g_0$, and $\theta$.**
>
>   Our model is based on the *single-index assumption*, namely that the conditional Fréchet mean depends on the predictors through a scalar projection, $g_0(X)=X^\top\theta_0$, where $\theta_0\in\mathbb{R}^p$ is the index direction. Thus, in the population model, $g_0$ is assumed to be linear. This is not meant to say that the DNN learns an arbitrary map; rather, the DNN module is used to *estimate the index direction* $\theta_0$ in a flexible, data-adaptive way. More specifically, the last hidden layer of the DNN outputs a vector in $\mathbb{R}^p$, which is then normalized to unit length and denoted as $\theta$. This vector is combined with the input $X$ to form the scalar index $Z=X^\top \theta$, which serves as the input to the local Fréchet regression step. The normalization $|\theta|=1$ is imposed for *identifiability*, since otherwise the scale of $\theta$ is not identifiable; a single index vector is commonly interpreted as a projection direction.  In the revision, we will clarify the roles of the DNN, $g_0$, and $\theta$ to avoid ambiguity. For the question regarding subject-specific $\theta$, we refer to our response to Reviewer gtTb for further clarification.
>
> - **Comparison with additive models.**
>
>   We will expand the related work section to include additive-model approaches. We also conducted an additional comparison with the additive model of Han et al. (2020) in the distribution setting with a quadratic link. Over 100 repeated runs, the MSPEs for the additive model are 0.7536 (0.0154), 0.8069 (0.0179), and 0.9699 (0.0236) for sample sizes 200, 500, and 1000, respectively. These are larger than those obtained by DeSI (e.g., 0.1300 at $n=200$; Table 1), which is expected since the simulation data are generated under a single-index mechanism.
>
> - **Robustness beyond the linear single-index setting.**
>
>   To investigate robustness, we considered a data-generating mechanism that departs from the single-index assumption. Specifically, we generated distributional outputs as normal distributions $Y_i=N(\mu_i,1)$, where $\mu_i=N(X_{i1}^2+X_{i2}^2+X_{i3}^2+X_{i4}^2,0.25)$, and the inputs $X_i$ were generated as in Section 5.3. This corresponds to an additive model with component functions $f_i(x)=x^2$ for $i=1,2,3,4$. We compared four settings: GFR with input $X_i$, GFR with input $X_i^2$, DeSI with input $X_i$, and DeSI with input $X_i^2$. The MSPEs based on 100 repeated runs are reported below.
>     |Method|$n=200$|$n=500$|$n=1000$|
>     |-|-:|-:|-:|
>     |GFR with input $X$| 0.5889 (0.0270)|0.5905 (0.0181)| 0.5872 (0.0252)|
>     |GFR with input $X^2$|0.0599 (0.0225)|0.0400 (0.0146)|0.0284 (0.0092)|
>     |DeSI with input $X$|0.3977 (0.0414)|0.3765 (0.0367)| 0.3810 (0.0329)|
>     |DeSI with input $X^2$|0.0731 (0.0230)|0.0491 (0.0211)|0.0439 (0.0149)|
>   When using $X^2$ as input, GFR is correctly specified and consequently achieves the best overall performance, as expected. Under this ideal setting, DeSI does not outperform GFR. In contrast, when both methods are applied to the original input $X$, DeSI consistently outperforms GFR, suggesting greater robustness when the underlying model structure is misspecified. We will include these results in the revised manuscript to further illustrate the behavior of DeSI beyond the single-index setting.

---

> > ### Author Rebuttal · Reviewer_Q3xz · 2026-04-03
> >
> > The authors have clarified things that where previously not clear and I expect them to add those clarifications to the paper. They also added a comparison to additive models which the submitted paper was lagging.

---

### Decision · Program_Chairs · 2026-04-30

**Decision:**

Accept (regular)

**Comment:**

The reviewers are in consensus that this paper lies above the acceptance threshold, and the rebuttals successfully addressed all issues pointed out in the reviews. I therefore recommend acceptance.